# scLong: a billion-parameter foundation model for capturing long-range gene context in single-cell transcriptomics

Ding Bai [1,12], Shentong Mo [1,12], Ruiyi Zhang [2,12], Yingtao Luo [3], Jiahao Gao [4], Jeremy Parker Yang [5], Qiuyang Wu [6], Hamidreza Rahmani [7], Tiffany Amariuta [4,8], Danielle Grotjahn [7], Sheng Zhong [6], Nathan Lewis [6,9], Wei Wang [5,10], Trey Ideker [4,6], Pengtao Xie [1,2,4,6,8,11] ✉ & Eric Xing [1,3] ✉

Single-cell RNA sequencing (scRNA-seq) has revolutionized the study of cellular heterogeneity by providing gene expression data at single-cell resolution, uncovering insights into rare cell populations, cell-cell interactions, and gene regulation. Foundation models pretrained on large-scale scRNA-seq datasets have shown great promise in analyzing such data, but existing approaches are often limited to modeling a small subset of highly expressed genes and lack the integration of external gene-specific knowledge. To address these limitations, we present scLong, a billion-parameter foundation model pretrained on 48 million cells. scLong performs self-attention across the entire set of 28,000 genes in the human genome. This enables the model to capture long-range dependencies between all genes, including lowly expressed ones (containing unexpressed genes with zero expressions), which often play critical roles in cellular processes but are typically excluded by existing foundation models. Additionally, scLong integrates gene knowledge from the Gene Ontology using a graph convolutional network, enriching its contextual understanding of gene functions and relationships. In extensive evaluations, scLong surpasses both state-of-the-art scRNA-seq foundation models and task-specific models across diverse tasks, including predicting transcriptional responses to genetic and chemical perturbations, forecasting cancer drug responses, and inferring gene regulatory networks.

Single-cell transcriptomics enables the study of gene expression at the individual cell level, offering insights into cellular heterogeneity that bulk methods cannot reveal[1–3]. It allows for the identification of rare cell populations[4], uncovers cell-cell interactions[5], and provides a detailed map of gene regulation[6], making it an essential tool for advancing personalized medicine, drug discovery, and understanding cellular diversity. Foundation models have shown great promise in analyzing single-cell transcriptomics data[7–11]. Pretrained on large-scale

[1]Mohamed bin Zayed University of Artificial Intelligence, Masdar City, Abu Dhabi, UAE. [2]Department of Electrical and Computer Engineering, University of California San Diego, La Jolla, CA, USA. [3]Machine Learning Department, School of Computer Science, Carnegie Mellon University, Pittsburgh, PA, USA. [4]Department of Medicine, University of California San Diego, La Jolla, CA, USA. [5]Department of Chemistry and Biochemistry, University of California San Diego, La Jolla, CA, USA. [6]Department of Bioengineering, University of California San Diego, La Jolla, CA, USA. [7]Department of Integrative Structural and Computational Biology, The Scripps Research Institute, La Jolla, CA, USA. [8]Halıcıoğlu Data Science Institute, University of California San Diego, La Jolla, CA, USA. [9]Department of Pediatrics, School of Medicine, University of California San Diego, La Jolla, CA, USA. [10]Department of Cellular and Molecular Medicine, School of Medicine, University of California San Diego, La Jolla, CA, USA. [11]School of Biological Sciences, University of California San Diego, La Jolla, CA, USA. [12]These authors contributed equally: Ding Bai, Shentong Mo, Ruiyi Zhang. ✉e-mail: p1xie@ucsd.edu; epxing@cs.cmu.edu

single-cell RNA sequencing (scRNA-seq) datasets using self-supervised learning[12], these models can capture complex gene expression patterns across diverse cell types. One of the key mechanisms in these models is self-attention[13], which computes relationships between genes by allowing every gene to attend to every other gene. This helps the model capture important gene interactions, contextualize gene expression, and understand long-range dependencies between genes. With fine-tuning, foundation models can be adapted for various downstream tasks, such as cell type classification[7], gene perturbation prediction[14], and reconstruction of gene regulatory networks (GRNs)[15], even in settings with limited data.

Despite the significant progress foundation models have made in analyzing single-cell transcriptomics data, they still face critical limitations that hinder their ability to fully capture the complexity of gene expression. One key limitation is that, to save computational cost, these models typically perform self-attention on a small subset of genes (e.g., 2048 in Geneformer[8] and scFoundation[10], 2000 in scGPT[9], and 1536 in UCE[16]), often selected based on high expression levels[8]. This approach excludes many lowly expressed genes that play essential roles in cellular processes and regulatory networks[17–19]. By restricting self-attention to only a fraction of the transcriptome, current models miss important regulatory signals and fail to capture long-range gene interactions across the entire genome[20,21]. As a result, they provide incomplete representations of gene regulatory mechanisms, overlooking subtle but critical gene interactions that are key to understanding complex cellular functions. Another limitation is the lack of integration of external gene-specific knowledge, such as that provided by the Gene Ontology (GO)[22], which encodes relationships among genes, biological processes, and molecular functions. Current models[8–10] rely predominantly on patterns derived solely from gene expression data, which restricts their ability to capture context related to gene functions and regulatory interactions. Without leveraging such rich functional information, these models may struggle to fully understand the roles of genes, especially in cases where direct expression data offers limited insight into a gene's activity within a broader regulatory framework.

To overcome these limitations, we present scLong, a billion-parameter scRNA-seq foundation model. First, instead of focusing on a small subset of genes, scLong performs self-attention across the entire human transcriptome, encompassing around 28,000 genes. This enables the model to capture long-range interactions and dependencies between all genes, including those with low expression levels that may still play crucial roles in cellular processes. By including every gene in the analysis, scLong offers a more comprehensive and unbiased representation of GRNs, avoiding the pitfalls of restricting attention to highly expressed genes. Second, scLong integrates external gene knowledge from the GO using a graph convolutional network (GCN)[14,23] to learn gene representations. This allows the model to incorporate hierarchical and functional relationships between genes, providing deeper functional context to its predictions. By leveraging this structured information, scLong enhances its ability to capture gene functions and interactions, even when direct expression data is sparse or ambiguous. Together, these two mechanisms—self-attention across all genes and integration of GO knowledge—enable scLong to generate more accurate and functionally relevant representations, effectively addressing the limitations of current foundation models in transcriptomics data analysis. scLong has one billion parameters, making it substantially larger than scFoundation[10], GeneCompass[11] and UCE[16], which have 100 million, 100 million and 650 million parameters, respectively.

## Results
### scLong overview
scLong takes a cell's gene expression vector as input, generating a representation for each element in the vector (Fig. 1a). Each element

corresponds to a specific gene, with its value indicating the level of gene transcription into RNA at a given moment, which may reflect potential protein production. scLong includes a gene encoder, an expression encoder, and a contextual encoder. The expression encoder, a multi-layer perceptron (MLP), produces a representation vector for each scalar expression value. The gene encoder leverages GO[22] to extract a representation vector for each gene. For each element in the expression vector—defined by a gene ID and its expression value—we combine the gene's representation (from the gene encoder) with its expression representation (from the expression encoder) to represent the element. These element representations are then fed into the contextual encoder, which learns contextualized representations that capture relationships among elements (Methods).

The gene encoder constructs a gene graph using the GO and applies a GCN[14,23] to this graph to learn gene representations. The GO[22] offers a structured vocabulary for describing gene functions, organized into three primary domains: Biological Process, which refers to the biological roles or processes in which a gene is involved, such as cell division or metabolic pathways; Molecular Function, which specifies the biochemical activities of a gene product, such as enzyme activity or binding; and Cellular Component, indicating the cellular locations where a gene product operates, such as the nucleus or mitochondria. Each gene's functions are annotated with GO terms from this vocabulary. The gene graph is constructed based on the method in ref. 14, where each node represents a gene. For each pair of genes, $u$ and $v$, the Jaccard index is calculated to measure the overlap between their sets of annotated GO terms. If the overlap is sufficiently high, an edge is added between the two genes in the graph. The gene graph captures functional relationships between genes based on shared GO annotations. Genes with overlapping GO terms are connected, reflecting similarities in biological processes, molecular functions, and cellular localization. For example, genes involved in related biological processes, such as metabolic pathways, are linked, suggesting shared roles in complex cellular functions. Genes with similar molecular functions, like enzymatic activities or binding properties, are also connected, indicating biochemical similarities or cooperative interactions. Additionally, genes localized to the same cellular components, such as the nucleus or mitochondria, are linked, suggesting potential spatial co-localization. On top of the gene graph, we construct a GCN[24], which learns representations for each gene. Through a process called message passing, the GCN enables each node to aggregate information from its neighboring nodes, effectively capturing the relationships between genes.

The contextual encoder employs self-attention[13] to capture long-range relationships between genes in the context of the input cell. It takes the initial representations generated by the gene and expression encoders and learns a contextualized representation for each element. Self-attention calculates pairwise correlations among elements, capturing their interdependencies. To balance computational efficiency with representation quality, we use a large Performer[25] encoder and a mini Performer encoder to process elements with varying expression levels. Specifically, we rank each cell's gene expression elements in descending order, dividing them into two groups: a high-expression group, containing the top-ranked elements, and a low-expression group with the remaining ones. The high-expression group, which carries core biological information critical for modeling gene interactions and regulatory pathways, is processed by the larger Performer encoder with more layers and parameters. The low-expression group, offering less critical information, is processed by the smaller encoder, optimizing computational efficiency.

While low-expression genes (LEGs) are less prominent in terms of overall abundance, they play essential roles in a range of biological processes and cannot be disregarded. Many LEGs are involved in regulatory mechanisms that influence the behavior of high-expression genes, acting as switches or modulators in complex cellular networks[17]. These genes

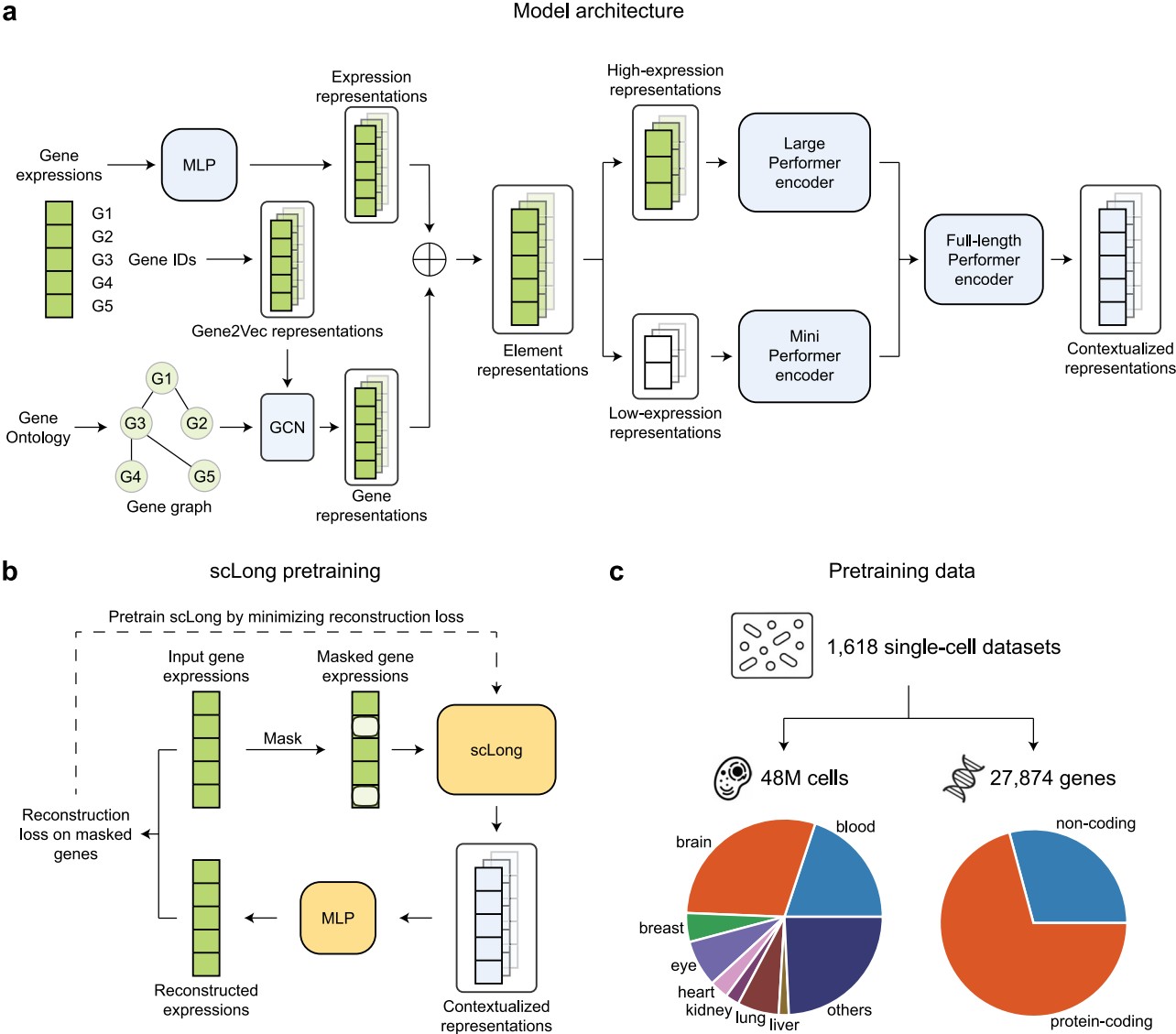

**Fig. 1 | scLong, a scRNA-seq foundation model with one billion parameters pretrained on 48 million cells, captures long-range context across 27,874 genes by employing a dual encoder architecture and leveraging Gene Ontology knowledge. a** Model architecture of scLong. scLong generates a representation for each element in a cell's gene expression vector using three main components: a gene encoder, an expression encoder, and a contextual encoder. The expression encoder, a multi-layer perceptron (MLP), produces a representation vector for each scalar expression value, while the gene encoder utilizes Gene Ontology to derive a representation vector for each gene. These representations are combined for each element and fed into the contextual encoder, which learns context-aware representations that capture inter-element relationships. Specifically, the gene encoder constructs a gene graph from Gene Ontology and applies a graph convolutional network (GCN) to learn gene-specific representations. To capture long-range relationships between genes, the contextual encoder leverages self-attention. To optimize efficiency and representation quality, scLong employs two Performers of different sizes, with high-expression elements processed by a larger Performer for detailed interaction modeling, and low-expression elements by a smaller Performer for efficiency. The outputs from these two encoders are then passed through a final full-length Performer, generating the final scLong representations. **b** scLong is pretrained by reconstructing masked expression values. For each input cell, we randomly mask a subset of expression values and use scLong to learn representations for both the masked and unmasked elements. The representations of the masked elements are passed to an MLP-based decoder to predict their expression values. A reconstruction loss is calculated between the predicted and actual values, and pretraining involves minimizing this reconstruction loss. **c** The pretraining data for scLong includes 48 million cells and 27,874 genes (~20,000 protein-coding and 8000 non-coding genes) derived from 1,618 scRNA-seq datasets spanning over 50 tissues.

can also be crucial in rare or specialized cell types, where their subtle expression may drive specific phenotypes or responses to environmental stimuli[18,19]. Ignoring them could lead to incomplete models that overlook important aspects of cellular function. Moreover, LEGs often participate in context-specific pathways that become active only under certain conditions, such as stress responses, immune signaling, or disease progression[18,19]. These genes may also be important for rare cell populations, whose contributions to tissue function or disease states could be missed if low-expression signals are not adequately represented[20,21]. Thus, while high-expression genes often drive primary biological processes, LEGs provide the fine-tuned regulation and specialized functions necessary for a complete understanding of cellular behavior.

After processing by the large and mini Performer encoders, each element obtains a contextualized representation vector of uniform dimension. These vectors are then input into a full-length Performer encoder, which performs self-attention across all elements. This final encoder, configured with the same number of layers as the mini encoder, produces the final representations for each expression element.

To pretrain scLong, we compiled a large-scale scRNA-seq dataset comprising ~48 million human cells from diverse tissues and cell types

(Fig. 1c), covering 27,874 human genes (Methods). Pretraining involves reconstructing masked expression values[12] (Fig. 1b). For each input cell, we randomly mask a subset of values, then use scLong to learn representations for both masked and unmasked elements. The representations of masked elements are fed into a decoder to predict their expression values. A reconstruction loss is calculated between the predicted and actual values. Pretraining is performed by minimizing these reconstruction losses (Methods).

## scLong predicts transcriptional outcomes of genetic perturbations

Predicting transcriptional outcomes of genetic perturbations involves forecasting how changes to specific genes, such as knockouts or over-expressions, impact the overall gene expression profile of a cell[26–28]. This capability is essential for understanding gene function and regulatory networks, as each perturbation can reveal how genes interact with each other and contribute to cellular behavior. Accurately predicting transcriptional outcomes offers a deeper understanding of pathways associated with disease, helping to pinpoint potential therapeutic targets and advance precision medicine. Additionally, in synthetic biology, understanding transcriptional responses supports the design of gene circuits and engineered cells with specific desired properties.

For this task, the input comprises a cell's pre-perturbation gene expression vector and its corresponding perturbation conditions, while the output is the cell's post-perturbation gene expression vector. We utilized scLong to generate representations for pre-perturbation gene expressions and employed GEARS[14] to derive representations for the perturbation conditions (Fig. 2a). These representations were summed and processed by a GEARS decoder to predict the post-perturbation gene expression vector (Methods). The perturbation conditions included both single and double gene perturbations, where either one or two genes were altered simultaneously in each cell. The Norman dataset[27], consisting of 91,205 cell samples, 5045 genes, and 236 unique perturbation conditions, was used for this task, providing a training set of 58,134 cells, a validation set of 6792 cells, and a test set of 26,279 cells. Each test sample was categorized into one of four scenarios: (1) neither gene in a double-gene perturbation conducted on the test sample is present in the training data (Seen 0/2); (2) one gene in a double-gene perturbation is absent from the training data (Seen 1/2); (3) both genes in a double-gene perturbation are present in the training data (Seen 2/2); and (4) the gene in a single-gene perturbation is absent from the training data (Seen 0/1). This categorization helps assess the model's ability to generalize to unseen perturbations. Following GEARS, prediction performance was evaluated using two metrics: Pearson correlation and mean squared error (MSE) on the top 20 differentially expressed (DE) genes[14] (Methods). We compared scLong with four state-of-the-art scRNA-seq foundation models: Geneformer[8], scGPT[9], scFoundation[10], UCE[16]. Geneformer, pretrained on 29.9 million cells, has 30 million parameters. scFoundation, with 100 million parameters, was pretrained on 50 million cells. scGPT, with 50 million parameters, was pretrained on 33 million cells. UCE, with 650 million parameters, was pretrained on 36 million cells. In addition, we compared scLong with task-specific approaches for predicting gene expression outcomes following genetic perturbations, including the deep neural network GEARS[14] and an additive linear model (ALM)[28] (Methods), both of which do not involve pretraining. Additionally, following[28], we included a simple baseline, *No-Change*, which directly uses the input gene expression vector as the predicted output.

scLong outperformed the seven baseline methods in most cases, across both Pearson correlation and MSE metrics, and under various test scenarios, including Seen 0/2, Seen 1/2, Seen 2/2, and Seen 0/1 (Fig. 2b and Supplementary Table 1; since the errors of ALM and No-Change are considerably larger than those of other baselines, we excluded them from Fig. 2 and instead provided their results in Supplementary Table 1). The improvement of scLong was particularly notable in the Seen 0/1 and Seen 0/2 scenarios, where the perturbation conditions in the test data are not encountered during training. For example, in the Seen 0/1 scenario, scLong achieved a Pearson correlation of 0.625, compared to 0.561 ($P = 0.001$, two-sided $t$-test after the Benjamini-Hochberg procedure of multiple hypothesis correction[29]; the detailed results of the two-sided $t$-test, with sample size = 5 and most effect sizes >1, are provided in Supplementary Table 11), 0.576 ($P = 0.002$), 0.577 ($P = 0.002$), 0.581 ($P = 0.002$), 0.530 ($P < 0.001$), 0.509 ($P < 0.001$), and $-0.012$ ($P < 0.001$) for GEARS, Geneformer, scGPT, scFoundation, UCE, ALM, and No-Change, respectively. In the Seen 0/2 scenario, scLong obtained an MSE of 0.170, while the baseline models recorded errors of 0.218 ($P = 0.001$), 0.185 ($P = 0.107$), 0.199 ($P = 0.005$), 0.190 ($P = 0.005$), 0.276 ($P < 0.001$), 0.346 ($P < 0.001$), and 0.503 ($P < 0.001$), respectively. This demonstrates that scLong has a stronger out-of-domain generalization capability compared to the baseline models.

We evaluated scLong's capability to classify double-gene perturbations into two genetic interaction (GI) types: synergy and suppressor. Following[27], we used a magnitude score to distinguish these interaction types. This score measures the correlation between the effect of a double-gene perturbation ($i, j$) and the linear combination of the effects of the corresponding single-gene perturbations $i$ and $j$ (Methods). Higher scores indicate stronger synergy between $i$ and $j$. For each double-gene perturbation, we calculated magnitude scores using ground-truth post-perturbation expressions, scLong-predicted post-perturbation expressions, and GEARS-predicted post-perturbation expressions. We then computed Pearson correlation coefficients (PCC) between scLong and the ground truth, as well as between GEARS and the ground truth. scLong achieved a higher PCC than GEARS (Fig. 2c), demonstrating its superior ability to identify the true GI type. To further illustrate this, we ranked the magnitude scores of ground truth, scLong, and GEARS in descending order. The top 15 and bottom 15 double-gene perturbations were classified as having synergy and suppressor GI types, respectively. For both interaction types, the overlap between scLong and the ground truth exceeded that of GEARS and the ground truth (Fig. 2d), further demonstrating that scLong more accurately predicts synergistic and suppressive gene interactions in double-gene perturbations.

Figure 2e shows scLong's mean absolute prediction errors for individual genes (columns) across various perturbation conditions (rows). We visualized 90 genes and 40 perturbation conditions with the highest prediction errors. Hierarchical clustering of perturbation conditions was performed, grouping those with similar error patterns (row vectors). The clustering results are sensible: conditions involving the same gene, such as CEBPB + LYL1 and PTPN12 + CEBPB, or CNN1 + UBASH3A and CBL + CNN1, were grouped together. Under conditions involving CEBPE (e.g., ZC3HAV1 + CEBPE, PTPN12 + CEBPE), scLong achieved near-zero errors across all genes. This is because CEBPE has minimal regulatory influence on other genes; perturbing CEBPE did not markedly affect gene expression, which simplified the prediction of transcriptional changes. In contrast, conditions involving ZBTB10, SNAI1, or DLX2 exhibited notably higher prediction errors, as these genes exerted substantial regulatory influence on others. Perturbing them triggered significant transcriptomic shifts, posing a greater challenge for accurate prediction. Generally, prediction errors were low across most genes; however, errors for the HBZ gene were particularly high due to its sensitivity to regulatory effects from other genes. Perturbing these regulators substantially altered HBZ expression, making its post-perturbation state more challenging to predict.

## scLong predicts transcriptional outcomes of chemical perturbations

Beyond predicting genetic perturbations, we applied scLong to predict gene expression profiles in response to de novo chemical perturbations, which is crucial for drug discovery and personalized medicine[30].

 

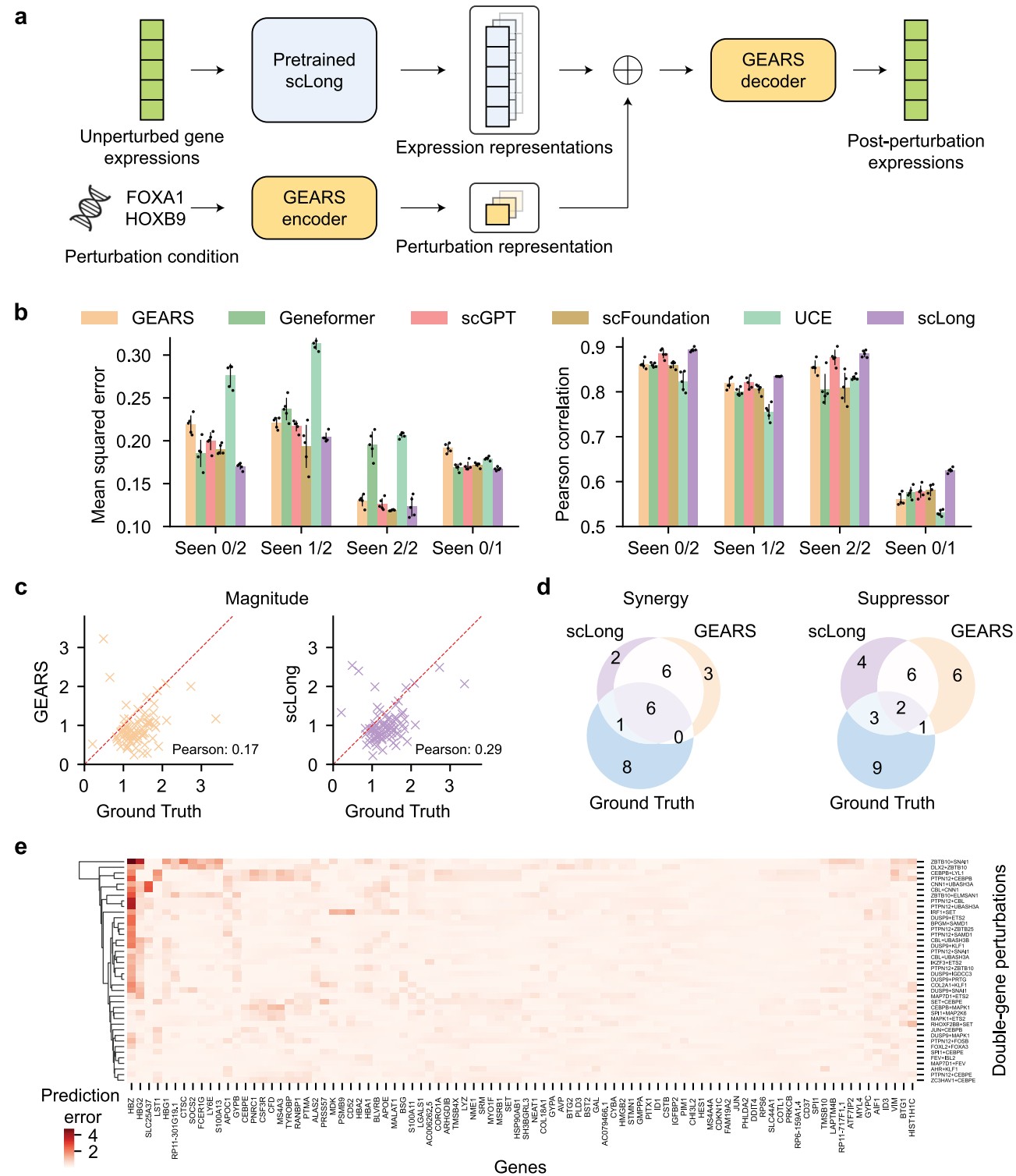

**b** GEARS · Geneformer · scGPT · scFoundation · UCE · scLong

**c** Magnitude
GEARS Pearson: 0.17
scLong Pearson: 0.29

**d** Synergy
scLong GEARS
Ground Truth

Suppressor
scLong GEARS
Ground Truth

**e** Prediction error

By forecasting how novel compounds affect gene activity, researchers can rapidly screen for potential therapeutic effects or adverse reactions, significantly accelerating the drug development process. This capability also provides insights into the molecular pathways and cellular processes targeted by new compounds, helping to uncover their specific mechanisms of action. Additionally, it reduces the need for extensive experimental validation, saving time and resources, while enabling more precise, data-driven decisions in both clinical and research settings.

In this task, we used a subset of the L1000 dataset[31], which contains 7 distinct cell lines, 978 genes, and 810 drug compounds, with

drugs tested at 6 different dosage levels. The prediction model takes two inputs: (1) the index of the perturbed cell line and (2) the molecular graph and dosage of the drug used to perturb it. The output is the gene expression profile of the cell line after perturbation. The dataset does not include pre-perturbation gene expression data. Each data sample in L1000 consists of these inputs and outputs, totaling 5005 examples, with 3965 used for training, 544 for validation, and 496 for testing. We used scLong to extract representation vectors for each gene and a GCN to extract representations from the drug molecule graph (Fig. 3a). These representations are passed through a multi-head cross-attention module[13], combined with embeddings of cell line indices and dosage

**Fig. 2 | scLong surpassed state-of-the-art scRNA-seq foundation models and task-specific methods in predicting transcriptional outcomes of genetic perturbations. a** Model architecture for fine-tuning the pretrained scLong to predict transcriptional outcomes of genetic perturbations. **b** scLong outperformed scRNA-seq foundation models, including Geneformer, scGPT, scFoundation and UCE, as well as the task-specific GEARS method, in terms of Pearson correlation (higher is better) and mean squared error (lower is better) on the top 20 differentially expressed genes across four testing scenarios: Seen 0/2, Seen 1/2, Seen 2/2, and Seen 0/1. **c** In classifying double-gene perturbations into two genetic interaction types, synergy and suppressor, scLong's magnitude score achieved a significantly higher Pearson correlation with the ground truth than GEARS, underscoring its enhanced capability to distinguish between these interaction types. Each cross denotes a double-gene perturbation. **d** The top 15 synergistic double-gene

perturbations identified by scLong showed a greater overlap with the ground truth compared to GEARS, and the same held for suppressor double-gene perturbations. This further demonstrated that scLong provides more accurate predictions of synergistic and suppressive interactions in double-gene perturbations. **e** scLong's mean absolute prediction errors for individual genes (columns) across different double-gene perturbation conditions (rows). The 90 genes and 40 conditions with the largest errors were visualized. Hierarchical clustering of error patterns (row vectors) effectively grouped perturbation conditions involving the same gene together. In (**b**), bar heights represent the mean and error bars indicate the standard deviation across $n = 5$ independent training runs with different random seeds. Results from individual runs are shown as dot points. Source data are provided as a Source Data file. Two-sided $t$-tests with Benjamini-Hochberg correction were used; see Supplementary Table 11 for detailed statistics.

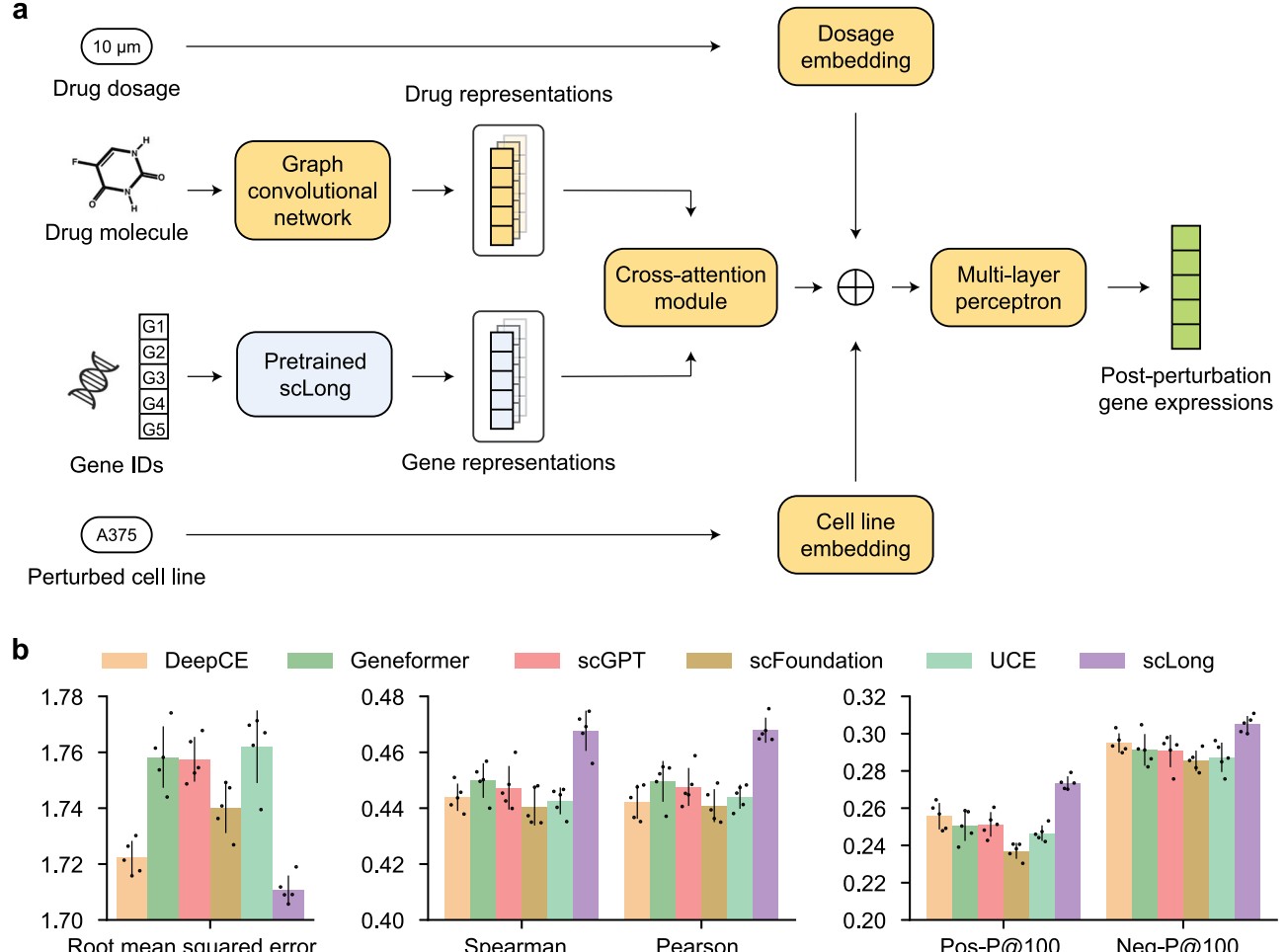

**Fig. 3 | scLong outperformed existing scRNA-seq foundation models and specialized methods in predicting transcriptional outcomes of chemical perturbations. a** Model architecture for fine-tuning the pretrained scLong for this prediction task. **b** scLong demonstrated superior results over scRNA-seq foundation models, including Geneformer, scGPT, scFoundation, and UCE, as well as the task-specific DeepCE method, across metrics including root mean squared error (RMSE), Spearman and Pearson correlations, Pos-P@100 and Neg-P@100. Higher values indicate better performance for all metrics except RMSE. In (**b**), bar heights represent the mean and error bars indicate the standard deviation across $n = 5$ independent training runs with different random seeds. Results from individual runs are shown as dot points. Source data are provided as a Source Data file. Two-sided $t$-tests with Benjamini-Hochberg correction were used; see Supplementary Table 12 for detailed statistics.

information, and then fed into an MLP to predict post-perturbation gene expression (Methods). We compared scLong with four foundation models, Geneformer, scGPT, scFoundation, and UCE, as well as the task-specific model DeepCE[32]. Evaluation metrics included root mean square error (RMSE), Spearman and Pearson correlation scores, and top-100 precision for the highest (Pos-P@100) and lowest (Neg-P@100) predicted expression values (Methods). For RMSE, lower values indicate better performance, while higher values are better for

the other metrics. scLong significantly outperformed all baseline methods across all evaluation metrics (Fig. 3b) (all $P < 0.04$, two-sided $t$-test with sample sizes of 5 and effect sizes greater than 2; see Supplementary Table 12).

### scLong predicts cancer drug response
Cancer drug response prediction involves forecasting how individual cancer cells or tumors will react to specific treatments[33]. This process is

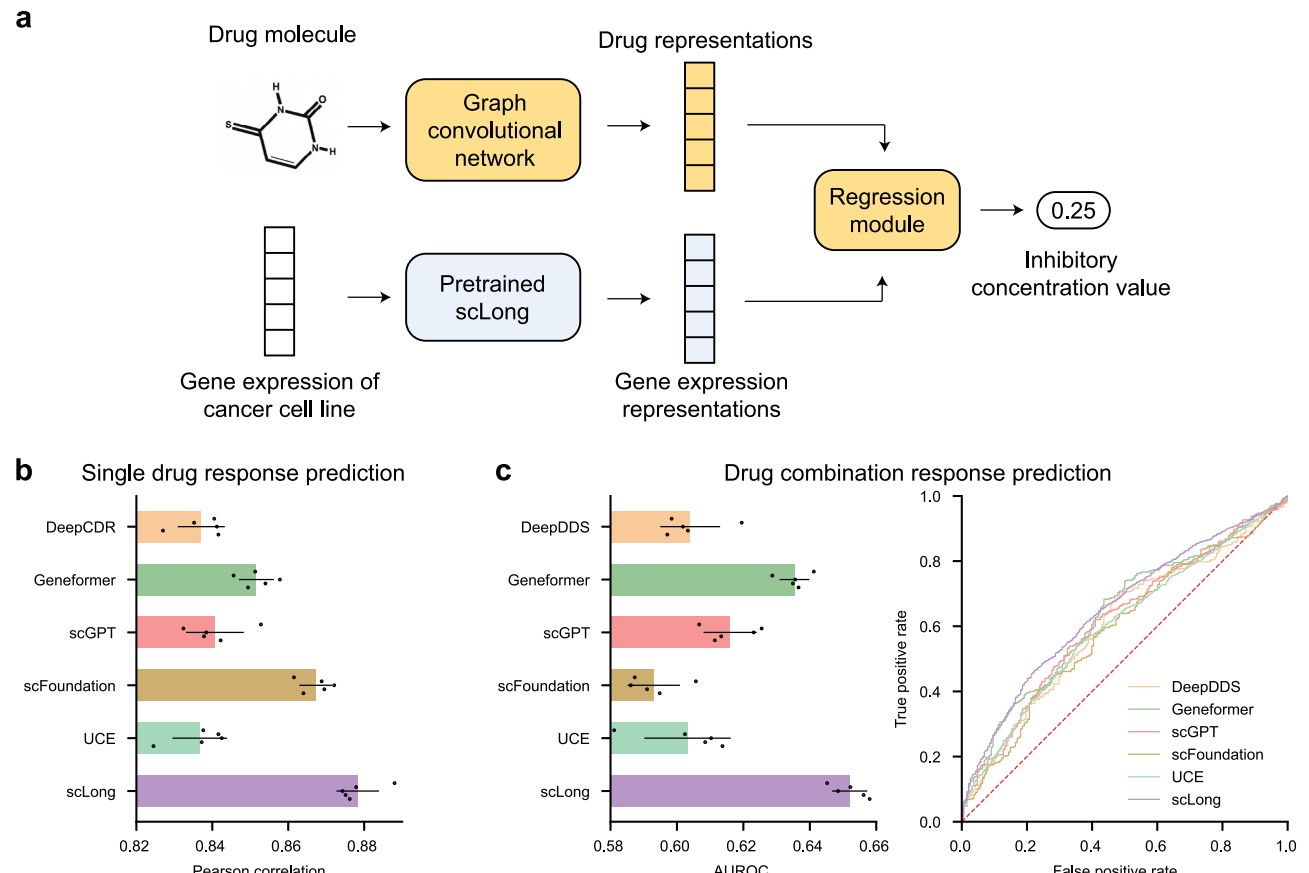

**Fig. 4 | scLong surpassed existing scRNA-seq foundation models and task-specific methods in predicting cancer cell responses to individual drugs and synergistic drug combinations. a** Model architecture for fine-tuning the pretrained scLong for this prediction task. **b, c** scLong achieved higher Pearson correlation and area under the receiver-operating characteristic curve (AUROC) than foundation models including Geneformer, scGPT, scFoundation and UCE, as well as specialized approaches including DeepCDR (**b**) and DeepDDS (**c**). In (**b, c**), bar heights represent the mean and error bars indicate the standard deviation across *n* = 5 independent training runs with different random seeds. Results from individual runs are shown as dot points. Source data are provided as a Source Data file. Two-sided *t*-tests with Benjamini-Hochberg correction were used; see Supplementary Table 13 for detailed statistics.

essential because cancer is a highly heterogeneous disease, and not all patients respond to the same drugs in the same way. By accurately predicting drug response, personalized treatment plans can be developed, improving the effectiveness of therapies and minimizing adverse effects. It enables oncologists to tailor treatment strategies based on the molecular profile of a patient's cancer, leading to better outcomes. Additionally, it accelerates drug discovery by identifying promising drug candidates and reducing the need for extensive clinical trials.

In this task, the input includes the molecular structure of a potential cancer drug and the bulk gene expression profile of a cancer cell line. The output is a prediction of the drug's efficacy against the cancer cell line, measured by its half-maximal inhibitory concentration (IC50) value[34]. We use scLong to extract a representation vector from the input gene expression data, which is then concatenated with the drug molecule representation obtained through a GCN[35] (Fig. 4a). The combined representation is subsequently fed into a regression module to predict the IC50 value (Methods). We used the dataset from DeepCDR[36], which includes 102,074 training examples and 5,372 testing examples. We compared scLong with other foundation models, including Geneformer, scGPT, scFoundation, and UCE. Additionally, we evaluated its performance against task-specific models, including the deep neural network DeepCDR and a linear model[36,37]. Pearson correlation was used as the evaluation metric, where higher values indicate better performance. scLong outperformed all baselines

(Fig. 4b), with a Pearson correlation score of 0.878, surpassing Geneformer's score of 0.852 (*P* = 0.001), scGPT's 0.841 (*P* = 0.001), scFoundation's 0.867 (*P* = 0.025), UCE's 0.837 (*P* = 0.001), DeepCDR's 0.837 (*P* = 0.001), and the linear model's 0.746 (*P* < 0.001) (Supplementary Table 3).

We further explored whether scLong improves the prediction of cancer cell responses to synergistic drug combinations, focusing on the response to drug pairs rather than individual drugs[38]. Drug combinations can target multiple pathways or mechanisms simultaneously, potentially leading to better therapeutic outcomes than single-drug treatments. They can also decrease the likelihood of drug resistance developing, as it is harder for cancer cells to adapt to multiple pharmacological agents at once. Despite its promise in cancer therapy, the exponential increase in potential drug pairings poses a significant challenge in identifying the most effective combinations. For this task, the input consists of a cancer cell line and a drug pair, while the output is a binary label indicating whether the cell responds. The model architecture closely resembles that used for single-drug response prediction (Methods). We used a large-scale oncology screening dataset[39] with over 12,000 examples for training and a separate dataset from AstraZeneca's drug combination dataset[40] with 668 samples for testing. The test dataset has a different distribution from the training data, allowing us to assess the models' out-of-distribution generalization capabilities. We compared scLong with Geneformer, scGPT, scFoundation, UCE, and task-specific models

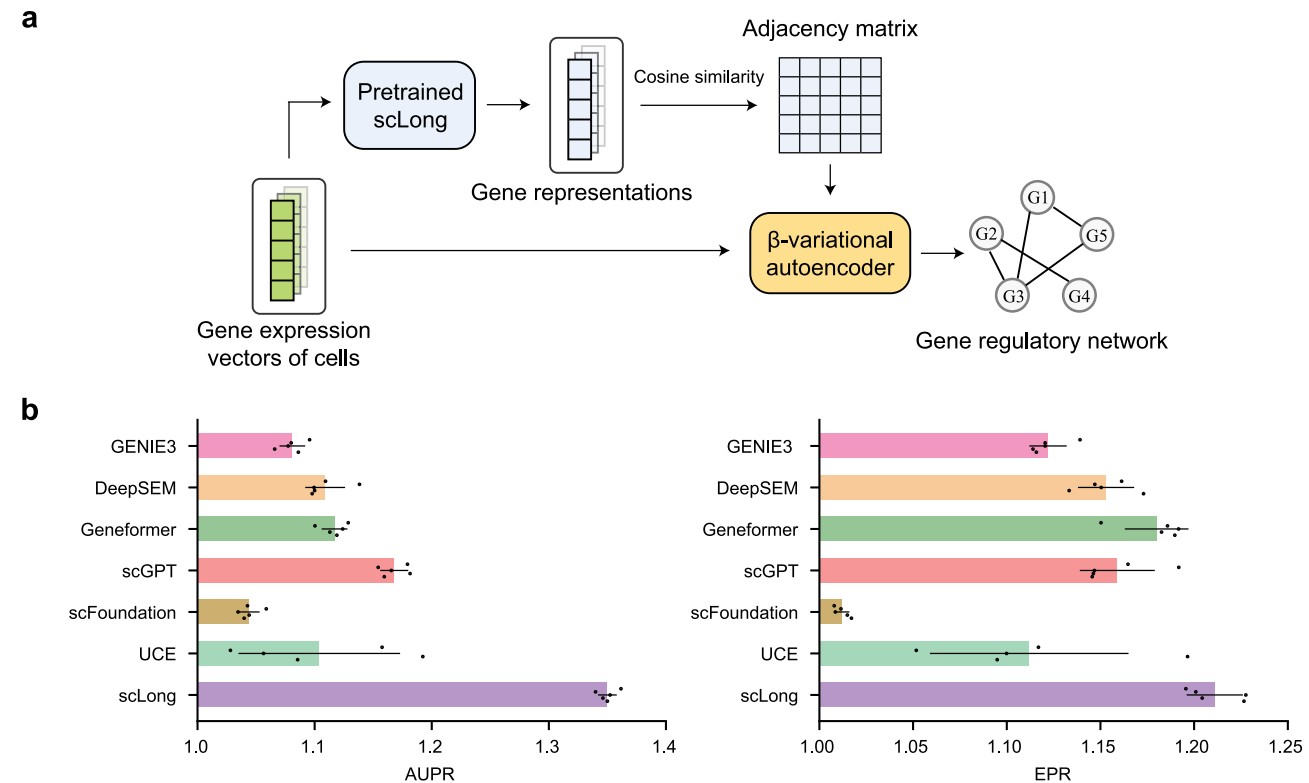

**Fig. 5 | scLong outperformed existing scRNA-seq foundation models and task-specific methods in gene regulatory network inference. a** Model architecture for fine-tuning the pretrained scLong for this inference task. **b** scLong achieves higher area under the precision-recall curve ratio (AUPR) and early precision ratio (EPR) compared to Geneformer, scGPT, scFoundation, UCE, and task-specific DeepSEM and GENIE3. In (**b**), bar heights represent the mean and error bars indicate the standard deviation across $n = 5$ independent training runs with different random seeds. Results from individual runs are shown as dot points. Source data are provided as a Source Data file. Two-sided $t$-tests with Benjamini-Hochberg correction were used; see Supplementary Table 14 for detailed statistics.

including DeepDDS[41] and random forest[37,41]. scLong outperformed all baselines in terms of the area under the receiver operating characteristic curve (AUROC) (Fig. 4c), with an AUROC score of 0.652, surpassing Geneformer's score of 0.635 ($P = 0.006$), scGPT's 0.616 ($P = 0.002$), scFoundation's 0.593 ($P < 0.001$), UCE's 0.603 ($P = 0.002$), DeepDDS's 0.604 ($P = 0.001$), and random forest's 0.533 ($P < 0.001$) (Supplementary Table 4). The results of the two-sided $t$-test after multiple hypothesis correction for both single-drug response prediction and drug-combination response prediction are presented in Supplementary Table 13, with a sample size of 5 for each test and an effect size greater than 2.

### scLong infers gene regulatory networks

GRNs represent the intricate interactions between genes and their regulators, such as transcription factors, that control gene expression within cells[42,43]. These networks determine which genes are turned on or off, guiding important cellular activities like differentiation, proliferation, and responses to environmental signals. Inferring GRNs from experimental data, such as scRNA-seq, is crucial for uncovering the regulatory mechanisms that drive these processes. Reconstructing these networks provides valuable insights into the molecular foundations of health and disease, highlighting key regulatory elements that may serve as potential therapeutic targets or biomarkers. Accurate GRN inference also enhances the ability to model and predict cellular behavior in response to specific conditions or treatments.

The input for this task consists of gene expression vectors from a collection of cells, with the output being a GRNs represented as an adjacency matrix. We used gene expression data from $N_c = 758$ human embryonic stem cells (hESC)[44,45], encompassing $N_g = 17,735$ genes. The

pretrained scLong model is applied to extract representations of these genes, and an adjacency matrix is generated by calculating cosine similarities between these gene representations (Fig. 5a). This matrix is further refined using the beta variational autoencoder[46] in DeepSEM[15] (Methods). We evaluated this GRN by comparing it to a ground-truth GRN derived from ChIP-Seq[47] data. Area under the precision-recall curve ratio (AUPR) and early precision ratio (EPR)[44], where higher values indicate better performance, were used as evaluation metrics (Methods).

We compared scLong's performance with Geneformer, scGPT, scFoundation, UCE, and task-specific methods including DeepSEM and GENIE3[48]. Additionally, we included a simple baseline, *GO Graph*, which directly utilizes the corresponding subgraph of the gene graph (Fig. 1a) derived from the GO as the GRN for the 17,735 genes. scLong outperformed all baselines across both metrics (Fig. 5b and Supplementary Table 5). For instance, scLong achieved an AUPR of 1.35, significantly surpassing Geneformer (1.12, $P < 0.001$), scGPT (1.17, $P < 0.001$), scFoundation (1.04, $P < 0.001$), UCE (1.10, $P = 0.001$), DeepSEM (1.11, $P < 0.001$), GENIE3 (1.08, $P < 0.001$), and GO Graph (1.02, $P < 0.001$) (Supplementary Table 5). These results indicate that scLong's learned representations effectively capture gene interactions. A two-sided $t$-test (Supplementary Table 14) confirmed the significance of these improvements ($P < 0.04$ for both AUPR and EPR comparisons after multiple hypothesis correction), with sample sizes of 5 and most effect sizes exceeding 2.

### scLong supports zero-shot batch integration

Single-cell transcriptomic datasets often exhibit batch effects[49], which are systematic variations in gene expression caused by differences in

experimental conditions, such as sample preparation, sequencing platforms, reagent batches, and handling procedures. A batch refers to a group of cells processed under the same conditions. These technical variations can obscure biological signals and lead to misleading conclusions if uncorrected. To address batch effects, batch integration methods[50–52] have been developed to adjust for these variations, enabling the harmonization of data from multiple experiments while preserving genuine biological differences.

We evaluated the zero-shot capability of scLong in batch integration, comparing it with foundation models including Geneformer, scGPT, scFoundation, and UCE. Zero-shot means that these models were neither pretrained nor fine-tuned on the dataset used for this task. Following[53], each foundation model was used to extract cell representations from different batches. The goal was to assess whether batch effects, present in the original gene expression vectors, were reduced or eliminated in the extracted representations. In scLong, cell representations were formed by concatenating contextualized representations with reconstructed expressions, followed by sum-pooling (Methods). We conducted this analysis using the widely used pancreas dataset[49], which consists of 19,093 genes and 16,382 cells from six batches collected with different scRNA-seq technologies. In addition to foundation models, we compared scLong with baselines from[53], including: (1) *Raw*, which directly uses the raw expression values without performing batch integration; (2) *HVG*[53], which selects the top 2000 highly variable genes (HVGs) and retains only their expression values for each cell; and (3) a task-specific method scVI[54], trained on the pancreas dataset. Batch integration performance was evaluated using the modified batch-wise average silhouette width (batch ASW)[49], where higher scores indicate better performance.

scLong achieved a batch ASW score of 0.96, markedly surpassing all baselines (Fig. 6), including Raw (0.70), HVG (0.89), scVI (0.85), Geneformer (0.62), scGPT (0.89), scFoundation (0.71), and UCE (0.83). These results indicate that the representations learned by scLong effectively mitigate batch effects. Notably, despite not being pretrained or fine-tuned on the pancreas dataset, scLong outperformed scVI, which was trained on this dataset, highlighting its strong zero-shot capability.

### Ablation studies of scLong demonstrate the effectiveness of modeling low-expression gene and incorporating the Gene Ontology graph

We conducted ablation studies to assess the contributions of two components in scLong, namely (1) modeling LEGs and (2) incorporating the GO graph. These studies were conducted on three downstream tasks: predicting transcriptional responses to genetic perturbations, inferring GRNs, and integrating batches. To evaluate the impact of modeling LEGs, we compared the full scLong model, which learns representations for both high-expression genes and LEGs, with two ablation settings (Methods). In the first setting, *w/o LEG*, we retained only the top 4,096 high-expression genes per cell, omitting LEGs. Consequently, the mini Performer encoder, responsible for learning representations of LEGs, was removed. In the second setting, *Random LEG*, the weight parameters of the mini Performer encoder were replaced with values randomly sampled from a normal distribution. Across the three downstream tasks, scLong outperformed both ablation settings in most cases (Fig. 7a). For example, in GRN inference, scLong achieved an AUPR of 1.35, exceeding the 1.11 AUPR of w/o LEG ($P < 0.001$) and the 1.10 AUPR of Random LEG ($P < 0.001$). Similarly, in batch integration, scLong attained a batch ASW of 0.96, compared to 0.88 and 0.90 for the ablation settings. These results underscore the benefits of incorporating LEGs into the model.

In the second ablation study evaluating the impact of incorporating the GO graph, we compared the full scLong model with two ablation settings (Methods). In the first setting, *w/o GO*, we removed the GO graph and the GCN used to learn gene representations. Instead,

Gene2Vec representations were directly used as the final gene representations. In the second setting, *Random GO*, we replaced the GO graph with a randomly generated graph. Across the three downstream tasks, scLong generally outperformed both ablation settings (Fig. 7b). For instance, in GRN inference, scLong achieved an AUPR of 1.35, exceeding the 1.12 AUPR of w/o GO ($P < 0.001$) and the 1.12 AUPR of Random GO ($P < 0.001$). Likewise, in batch integration, scLong achieved a batch ASW of 0.96, compared to 0.91 and 0.93 for the ablation settings. These findings demonstrate the importance of incorporating GO in learning gene representations.

### scLong clusters marker genes associated with different cell types

Figure 8 illustrates the clustering of gene representations obtained from scLong for two cell types in the Zheng68K dataset[55] (with other 9 cell types shown in Supplementary Fig. 3). For each cell type, we randomly sampled 50 cells, used scLong to extract their gene representations, computed pairwise cosine similarity between genes, and conducted hierarchical clustering on the resulting similarity matrix (Methods). Displayed are the 50 genes with the highest similarity scores. The results show that marker genes for each cell type, highlighted in red, were grouped into the same cluster. These cell-type-specific genes were generally highly expressed within their respective cell types. Marker genes and non-marker genes were assigned to separate clusters. These results demonstrate scLong's capability to capture gene co-expression patterns within specific cell types.

## Discussion
scLong offers a valuable advancement in single-cell transcriptomics, providing a foundation model that accommodates the full spectrum of gene expression within single cells. Its billion-parameter architecture and dual-encoder strategy allow it to handle both high- and LEGs, addressing limitations in existing models that often overlook LEGs critical for cellular regulation. By integrating gene-specific information from the GO, scLong brings contextual depth to its predictions, enhancing its ability to capture nuanced interactions across diverse cellular contexts. This comprehensive approach not only improves prediction accuracy but also broadens the model's applicability in studying condition-specific responses and complex gene regulatory mechanisms. These capabilities make scLong a valuable tool for advancing research in precision medicine, drug discovery, and cellular biology, supporting new insights into gene expression dynamics and informing more targeted therapeutic approaches.

In predicting transcriptional outcomes of genetic perturbations, scLong's superior performance over existing foundation models can be attributed to two key advantages: comprehensive self-attention across all genes and the integration of GO knowledge. First, scLong's self-attention spans all 28,000 genes, capturing interactions among both highly and lowly expressed genes, unlike baseline models that restrict attention to a small subset of highly expressed genes. Although LEGs are often less abundant, they play essential roles in gene regulation and cellular signaling, acting as modulators that influence how high-expression genes respond to perturbations[17–19]. By attending to all genes, scLong identifies a more complete picture of regulatory dynamics, capturing subtle but important gene interactions crucial for accurately predicting the transcriptional effects of genetic perturbations. This broad gene attention allows scLong to account for dependencies and feedback mechanisms that baseline models may overlook due to their limited gene focus. The effectiveness of modeling LEGs for predicting transcriptional outcomes of genetic perturbations was verified through ablation studies (Fig. 7a). Second, scLong incorporates GO knowledge through a GCN, providing each gene with a representation enriched by its biological functions, processes, and cellular roles. GO offers structured, hierarchical insights that enable the model to understand not only direct gene interactions but also the

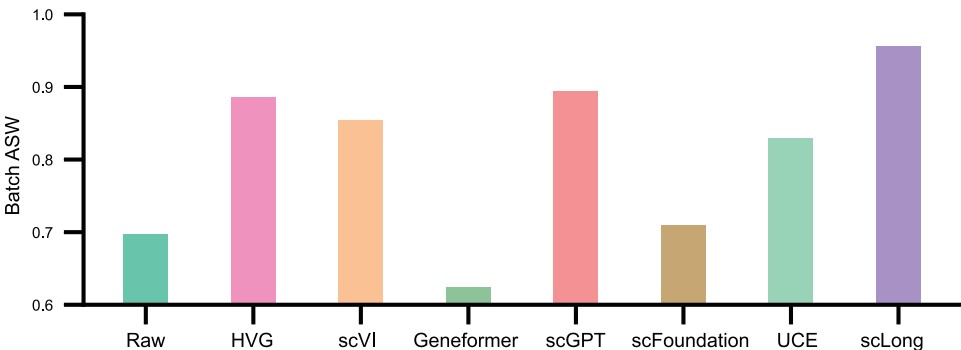

**Fig. 6 | scLong outperformed existing scRNA-seq foundation models and other baselines in zero-shot batch integration.** scLong achieved a higher batch average silhouette width (batch ASW) than Geneformer, scGPT, scFoundation, and UCE. It also outperformed the task-specific scVI method and two additional baselines, Raw and HVG. As this is a zero-shot task that does not involve fine-tuning, the results are deterministic and do not include error bars.

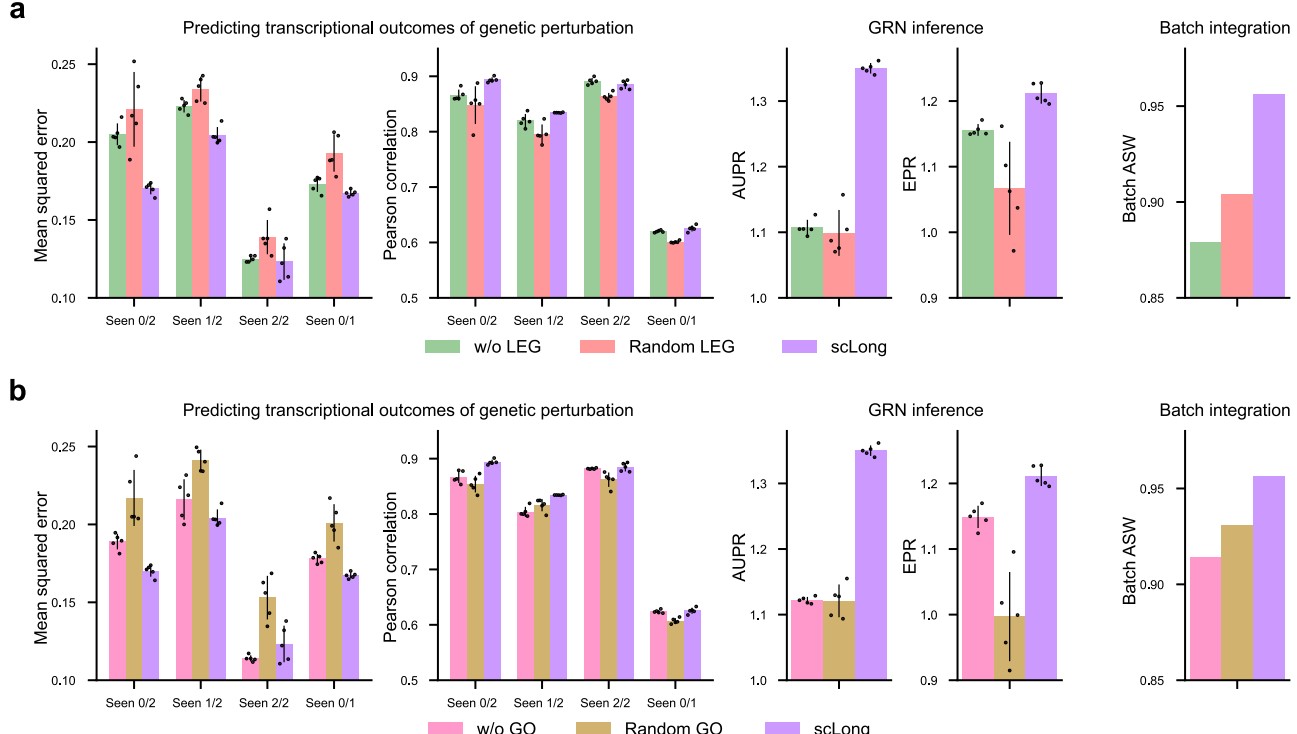

**Fig. 7 | Ablation studies of scLong demonstrate the benefits of modeling low-expression genes and integrating the Gene Ontology (GO) graph. a** In the ablation study assessing the impact of modeling low-expression genes (LEGs), the full scLong model significantly outperformed two ablation settings: (1) omitting LEGs and the mini Performer encoder (w/o LEG) and (2) assigning random weights to the mini Performer encoder (Random LEG). **b** In the ablation study evaluating the role of the GO graph, scLong significantly outperformed two ablation settings: (1) removing the GO graph and the graph convolutional network (w/o GO) and (2) replacing the GO graph with a random graph (Random GO). Higher values indicate better performance across all metrics, except for mean squared error. In (**a**, **b**), bar heights represent the mean and error bars indicate the standard deviation across $n = 5$ independent training runs with different random seeds. Results from individual runs are shown as dot points. Source data are provided as a Source Data file. Two-sided $t$-tests with Benjamini-Hochberg correction were used; see Supplementary Tables 15, 16 for detailed statistics.

broader functional context of each gene's role within the cellular environment[22,27,56]. In the context of genetic perturbations, such functional insights are vital, as they allow the model to infer how perturbing one gene might affect other related genes within the same biological pathways or processes. Baseline models that lack GO knowledge miss this critical functional layer. Ablation studies confirmed the effectiveness of incorporating the GO graph (Fig. 7b). Together, scLong's inclusive self-attention and GO-enhanced representations equip it to generate highly context-aware predictions, leading to its stronger performance in predicting transcriptional responses to genetic perturbations.

scLong's enhanced performance over the task-specific model GEARS in predicting transcriptional outcomes of genetic perturbations is largely due to its extensive pretraining on 48 million scRNA-seq samples—a foundational step that GEARS lacks. This large-scale pretraining enables scLong to learn generalizable patterns in gene expression across diverse cell types and conditions, equipping it with a comprehensive understanding of cellular behaviors, gene interactions, and regulatory networks. Genetic perturbations often result in complex regulatory cascades and cross-gene effects that are not fully represented within narrow task-specific datasets. Through exposure to tens of millions of expression profiles, scLong learns robust gene

**a**

Cell type: CD14+Monocyte

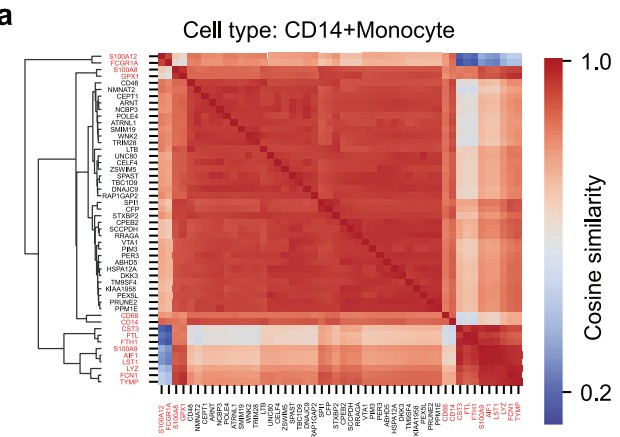

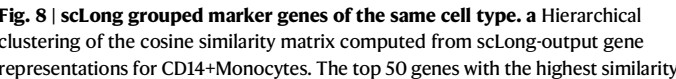

**b**

Cell type: CD19+B

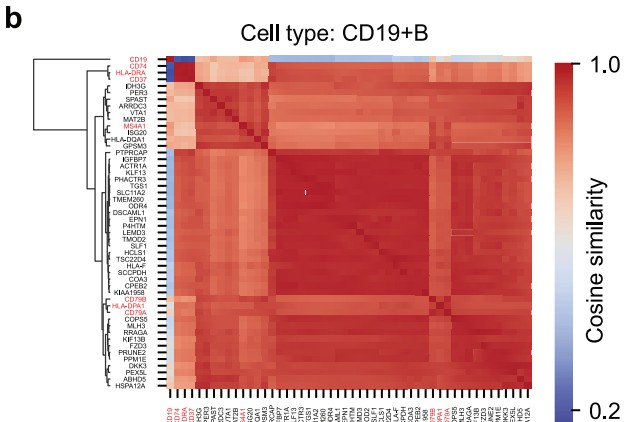

**Fig. 8 | scLong grouped marker genes of the same cell type. a** Hierarchical clustering of the cosine similarity matrix computed from scLong-output gene representations for CD14+Monocytes. The top 50 genes with the highest similarity scores are shown. Marker genes (highlighted in red) cluster together, while marker and non-marker genes are separated into distinct clusters. **b** The corresponding hierarchical clustering map for CD19+B cells.

representations that capture both common and rare expression patterns, including context-specific dependencies likely to be triggered by perturbations. This pretraining allows scLong to identify subtle transcriptional shifts and regulatory changes that may be pivotal in accurately predicting perturbation outcomes. In contrast, GEARS, lacking this extensive pretraining, relies solely on task-specific data, which limits its exposure to the broad spectrum of cellular states and gene regulatory mechanisms that scLong acquires through pretraining. Consequently, GEARS may struggle to capture nuanced gene interactions or transcriptional changes, particularly in response to less common or complex perturbations. Additionally, scLong's pretraining fosters the formation of robust gene representations, reducing overfitting to specific datasets and enhancing its predictive performance across diverse perturbation scenarios. This foundational knowledge enables scLong to make more accurate, context-aware predictions of transcriptional outcomes, leading to its stronger performance relative to GEARS in the task of genetic perturbation response prediction. Similarly, the superior performance of scLong over the ALM stems from its ability to learn comprehensive representations of both highly and lowly expressed genes, its integration of the GO for gene representation learning, and its extensive pretraining on a large-scale dataset. These features enable scLong to capture complex gene interactions and to learn generalizable patterns of expression across diverse conditions. This capability is especially important for accurately predicting transcriptional outcomes of genetic perturbations not present in the training data, resulting in substantial gains over the ALM, particularly in the Seen 0/2 and Seen 1/2 scenarios. A similar explanation accounts for scLong's advantage in predicting transcriptional outcomes of chemical perturbations, where it outperforms both existing foundation models and the task-specific model DeepCE.

In predicting cancer drug response, scLong's superior performance over existing foundation models is again attributed to its comprehensive self-attention across both highly and lowly expressed genes and its integration of GO knowledge. First, LEGs, while less abundant, often act as modulators within cellular pathways, serving as "switches" or fine-tuners in signaling and gene regulation that indirectly influence how high-expression genes respond to drug interventions[18,19,21]. In cancer cells, these subtle regulatory roles are particularly significant, as LEGs can control key pathways linked to drug resistance, cell survival, or proliferation[17,19–21]. By attending to LEGs, scLong captures a more complete view of the cellular network, identifying intricate dependencies and regulatory feedback loops that are essential for accurately predicting cancer cell responses to drug

interventions. Second, incorporating functional relationships from GO enriches scLong's predictions by embedding structured knowledge about gene functions and pathways. Since drugs often target or disrupt specific cellular processes, GO allows the model to recognize interactions among genes involved in critical processes like apoptosis, cell proliferation, and drug metabolism—central to a cancer cell's response to treatment. Additionally, GO annotations enable scLong to identify context-dependent roles of genes, including those that might be inactive under normal conditions but become essential under drug-induced stress. This integration of GO knowledge allows scLong to make more context-aware predictions, enhancing its ability to anticipate complex drug response patterns in cancer cells.

scLong's superior performance over task-specific models, including DeepCDR and DeepDDS, in predicting cancer drug response is again attributed to its extensive pretraining on 48 million scRNA-seq data points. Predicting cancer drug response requires understanding the complex interactions and regulatory networks that dictate how cancer cells respond to treatment[34]. This pretraining exposes scLong to a diverse range of gene expression profiles, enabling it to learn the underlying relationships and dependencies among genes, including those critical for modulating drug response in cancer cells. Additionally, gene expression patterns associated with drug resistance, metastasis, or apoptosis often appear only in specific cancer subtypes or under certain treatment conditions[34,57]. By capturing these rare but pivotal patterns, pretraining equips scLong to better understand and predict responses in heterogeneous cancer cell populations.

In GRN inference, scLong again outperforms baselines due to its self-attention mechanism operating across the entire set of ~28,000 genes, encompassing both highly and lowly expressed genes. This broad inclusion allows scLong to detect critical regulatory patterns that might be overlooked when focusing solely on highly expressed genes, as existing foundation models do. Lowly expressed genes play crucial roles in cellular regulation, including acting as fine-tuners in regulatory networks or contributing to rare but essential cellular processes[21]. By accounting for the expression patterns of these genes, scLong can infer a more complete and biologically accurate regulatory network. This comprehensive approach enables scLong to capture a broader range of gene interactions and dependencies, which are particularly important for uncovering regulatory relationships that influence rare or condition-specific cellular states. Ablation studies confirmed the value of including LEGs in GRN inference (Fig. 7a). Furthermore, scLong leverages the GO to construct functionally enriched representations of genes, adding another layer of precision in

regulatory inference. By incorporating GO, scLong is effectively preconditioned with structured knowledge about gene functions, pathways, and interrelations. This knowledge-rich representation provides a strong inductive bias, guiding scLong in recognizing functionally relevant connections between genes and their roles in regulatory networks. In contrast, existing foundation models lack this knowledge-based guidance, limiting its ability to identify meaningful relationships in cases where gene expression alone does not reveal functional interactions. The effectiveness of incorporating the GO graph was demonstrated through ablation studies (Fig. 7b). Furthermore, scLong outperforms the GO Graph baseline by a large margin (Supplementary Table 5), indicating that scLong does not merely replicate the GO graph but instead refines and extends its information during training. The GO-derived represents a cell type-agnostic view of gene interactions. In contrast, the task at hand requires inferring a GRN tailored to hESC using hESC-specific gene expression data. By incorporating such data, scLong is able to capture critical cell type-specific gene interaction patterns, leading to a more accurate GRN compared to the direct use of the GO graph.

In batch integration, scLong outperforms baselines for two key reasons. First, it mitigates batch effects by applying self-attention across the entire transcriptome, encompassing both high- and LEGs. By capturing long-range dependencies among all genes, scLong constructs a more comprehensive and biologically coherent representation of cells, reducing batch-specific artifacts. LEGs often participate in regulatory pathways that influence the expression of highly expressed genes. By integrating these regulatory interactions, scLong better distinguishes true biological variation from technical noise, resulting in more robust and transferable representations across batches. This comprehensive modeling of gene interactions helps preserve biological signals while minimizing batch-related discrepancies. Ablation studies demonstrated the benefit of incorporating LEGs in batch integration (Fig. 7a). Second, scLong enhances gene representations by incorporating the GO graph, which encodes hierarchical gene functions, molecular interactions, and pathway information. This structured knowledge enables the model to prioritize biologically meaningful signals over batch-specific artifacts. By leveraging functional relationships rather than relying solely on expression similarity, scLong aligns cell representations across datasets in a biologically consistent manner. This reduces overfitting to batch-specific noise and leads to more effective batch integration. Ablation studies confirmed the effectiveness of incorporating the GO graph (Fig. 7b).

scLong's ability to outperform both state-of-the-art foundation models for scRNA-seq and task-specific models across diverse downstream tasks underscores its robustness and adaptability in single-cell transcriptomics. Its strong performance in predicting transcriptional responses to genetic and chemical perturbations highlights its potential to aid in uncovering gene functions, regulatory pathways, and cellular responses under various conditions—essential for understanding disease mechanisms and identifying therapeutic targets. In the context of cancer research, scLong's accurate predictions of drug responses, both for individual drugs and synergistic drug combinations, present valuable insights for precision oncology. This capability could facilitate personalized treatment approaches by helping to identify the most effective therapies based on specific cancer cell profiles, potentially improving patient outcomes and reducing adverse effects. Additionally, scLong's success in GRN inference signifies its capacity to map complex interactions among genes, supporting efforts to model cellular processes and regulatory circuits more precisely.

Balancing computational efficiency and representation quality presents a fundamental challenge in large-scale models like scLong, where both attributes are crucial yet often in conflict. Achieving high-quality representations typically requires complex processing, such as applying self-attention across long expression vectors, to capture intricate gene relationships accurately. However, this comprehensive modeling approach incurs significant computational costs, as attention operations scale quadratically with the number of elements, making such methods infeasible for gene expression vectors with tens of thousands of elements. Strategies aimed at enhancing efficiency, such as reducing the number of layers, attention heads, or hidden dimensions, often sacrifice representation richness and granularity, limiting the model's ability to detect subtle yet significant gene interactions. Some approaches improve efficiency by shortening vector length, excluding LEGs under the assumption that they contribute less to primary cellular insights[9,10]. While this reduces memory and computational loads, it sacrifices the model's quality, as many LEGs are crucial for regulatory functions and context-specific cellular responses. scLong addresses this trade-off through a dual encoder strategy that selectively applies a larger encoder to high-expression genes, which typically convey more essential functional information, while a smaller encoder processes LEGs. This selective approach optimizes computational resources, allowing scLong to maintain high-quality representations for critical elements while managing efficiency. By retaining all genes in its representation and adjusting resource allocation appropriately, scLong preserves essential interactions among both high- and LEGs, achieving a balance between computational efficiency and comprehensive representation quality.

In our model, we treat all genes with zero expression values as potentially informative and explicitly model them. There are two main scenarios that can give rise to zero expression. First, the gene is expressed at a low level, but its expression is not detected due to limited sequencing depth. Second, the gene is not expressed in that particular cell. In both cases, the gene may carry valuable information for downstream analysis and should not be disregarded. In the first case, lowly expressed genes—those with non-zero but low expression levels—can play important biological roles despite their modest abundance, as discussed above. In the second case, genes that are not expressed (i.e., have true zero expression) can still provide important biological information in single-cell transcriptomics. The absence of expression is often cell-type specific and can help distinguish one cell population from another. Additionally, the repression of certain genes may reflect underlying regulatory programs or epigenetic states that define a cell's identity or function. In developmental and disease contexts, gene silencing can indicate key shifts in cellular states, such as differentiation, dedifferentiation, or pathological transitions. Thus, modeling non-expressed genes helps capture both the presence and absence of regulatory signals that shape cellular behavior. scLong employs a dedicated Performer encoder to model these lowly expressed and unexpressed genes, in contrast to previous foundation models that typically disregard these genes. In the first scenario, although assigning a zero value to a lowly expressed gene introduces some noise, it remains a reasonable approximation since the true expression level is, by definition, very low and close to zero. As a result, modeling these zero values can still reflect the underlying patterns of low expression. In scLong, representation vectors are learned for these zero values using MLPs, enabling the model to capture informative patterns.

Despite its advancements, scLong has certain limitations that merit consideration. The model's billion-parameter architecture, although optimized for efficiency, still demands significant computational resources for training and inference, which may hinder accessibility for groups lacking high-performance infrastructure. Additionally, scLong relies on static, predefined relationships from sources like the GO, which, while providing valuable contextual information, may restrict adaptability to dynamic gene interactions and condition-specific regulatory changes not represented in these databases. Another limitation is the potential sensitivity of scLong's performance to the choice of high- and low-expression gene thresholds in its dual encoder design; selecting these thresholds

inappropriately could lead to suboptimal representations, particularly in cell types with unusual gene expression distributions. Addressing these limitations could make scLong a more versatile and broadly applicable tool in single-cell transcriptomics research.

Future work on scLong can focus on several key areas to further enhance its capabilities and broaden its applications. One promising direction is the incorporation of additional biological datasets, such as pathway databases[58], protein-protein interaction networks[59], and epigenetic data[60], to enrich the context-awareness of the model and improve its ability to capture more complex regulatory mechanisms. Expanding the model's pretraining on diverse datasets from various species and tissues could also boost its generalizability across different biological contexts. Another area for improvement is model interpretability; future versions of scLong could integrate more advanced explainability techniques, such as attention-based visualization tools or saliency maps[61], to provide clearer insights into the gene interactions driving its predictions. Additionally, exploring methods to reduce the computational demands of training and deploying scLong, such as model pruning[62] or distillation[63], would make the model more accessible to a wider range of researchers. Furthermore, applying scLong to novel downstream tasks, such as predicting cell signaling pathways[5] or identifying gene interactions in rare cell populations, could further validate its versatility and expand its impact in single-cell biology.

Another direction for future development is to incorporate alternative sources of prior knowledge—beyond the GO—to improve gene representation learning. One complementary approach is exemplified by UCE[16], which uses embeddings from protein language models trained on amino acid sequences to capture structural and biochemical properties in an unsupervised manner. This strategy reduces dependence on curated annotations while offering broad coverage. In contrast, scLong encodes structured biological semantics derived from GO, capturing functional roles, cellular localization, and biological processes with curated precision. These two strategies are complementary: sequence-derived embeddings offer generalizability and annotation-independence, while ontology-based embeddings provide access to higher-order biological context that may be difficult to infer from sequence alone. A systematic comparison of these approaches, along with the development of hybrid models that integrate both forms of prior knowledge, constitutes a valuable direction for future research.

## Methods
### Collection and preprocessing of large-scale transcriptomics pretraining data
We collected scRNA-seq data from three public repositories: CELLxGENE[64–68], Cell Blast[69], and the Human Cell Atlas[70]. Initially, around 1600 datasets were downloaded, comprising over 60 million cells. We filtered out non-human datasets and excluded those containing fewer than 1000 genes. Additionally, datasets normalized using unknown methods were removed. After this filtering process, 848 datasets remained.

Next, we performed gene selection. First, we removed all non-human genes from the 848 datasets, leaving ~66,000 human genes. From these, we selected the top 20,000 genes with the highest number of non-zero entries across the datasets. Additionally, we included all 19,748 protein-coding genes[71], all 20,480 genes from the GO[22], and all 20,184 genes from Gene2Vec[72]. To eliminate duplicates, we mapped gene IDs from these different sources—represented as gene symbols or NCBI IDs—into a unified format based on Ensembl IDs. After removing duplicates, we obtained a final list of 27,874 unique genes.

For each cell, we created a 27,874-dimensional gene expression vector based on the 27,874 selected genes, where the $j$-th element represents the expression value of gene $j$ in that cell. If a gene was not expressed in the cell, its value was set to 0. A cell was removed if it had

fewer than 300 non-zero expression values. We then checked whether the expression values were in raw counts or already log1p normalized[73]. If an expression value $x$ was in raw count, we applied log1p normalization to it: $x \leftarrow \log(x/10000 + 1)$. Next, we adjusted the normalized expression values by magnifying or clipping them so that the maximum value in each cell's expression vector was 10. If the maximum value in an expression vector exceeded 10, all values greater than 10 were set to 10. If the maximum value was less than 10, each value in the vector was scaled by dividing it by the maximum value and then multiplying by 10. After removing duplicate cells, we retained 48,024,242 unique cells.

In the pretraining dataset, 9.37% of the 48 million cells contain non-zero LEGs, amounting to ~4.5 million cells. Each of these cells has, on average, 2514 non-zero LEGs. Please refer to Supplementary Section 1 for details.

### scLong model architecture
Each element in a gene expression vector contains two components: the gene ID and its expression value. scLong employs a gene encoder to generate a representation vector for the gene and an expression encoder to produce a representation vector for the expression value. The final representation for each element is obtained by adding these two vectors. The expression encoder is a MLP that takes the scalar expression value as input and outputs a representation vector. This MLP consists of two layers, with ReLU activation[74], and generates a representation vector with a dimension of 200.

The gene encoder first uses Gene2vec[72] to obtain an initial 200-dimensional representation for each gene. It then constructs a gene graph based on the GO, which, along with the initial representations, is input into a GCN[24] to learn a refined representation for each gene. The gene graph is constructed as follows[14]: for each gene pair, $u$ and $v$, we retrieve their annotated GO terms from the GO, denoted as $N_u$ and $N_v$. GO terms are standardized categories that describe various attributes of genes, focusing on their roles, processes, and locations within a cell. They are organized into three main categories: Molecular Function, Biological Process, and Cellular Component. Each gene can be associated with multiple GO terms, providing a comprehensive view of its functional and spatial characteristics within cellular and molecular systems. We then compute the Jaccard index $J_{u,v} = \frac{|N_u \cap N_v|}{|N_u \cup N_v|}$ between the two sets of GO terms, which quantifies the fraction of shared GO terms and indicates the functional similarity of each gene pair. Using this similarity measure, we construct a graph where each gene is represented as a node, and edges are assigned between gene pairs with high Jaccard index values. Specifically, for each gene $u$, we select the top 20 genes $v_i$ with the highest $J_{u,v_i}$ values and connect them to $u$.

A one-layer GCN is constructed on the gene graph, taking $\mathbf{X}$, a matrix containing initial representation vectors generated by Gene2vec, as input. This GCN learns refined 200-dimensional representations $\mathbf{X}'$ for all genes using the following update equation:

$$\mathbf{X}' = \mathbf{D}^{-\frac{1}{2}}\widehat{\mathbf{A}}\mathbf{D}^{-\frac{1}{2}}\mathbf{X}\mathbf{\Theta}, \qquad (1)$$

where $\mathbf{\Theta}$ contains the GCN's weight parameters. Here, $\widehat{\mathbf{A}} = \mathbf{A} + \mathbf{I}$, with $\mathbf{A}$ representing the adjacency matrix of the gene graph and $\mathbf{I}$ as the identity matrix. $\mathbf{D}$ is a diagonal matrix with entries $D_{ii} = \sum_{j=1}^{K} \widehat{A}_{ij}$, where $K$ is the total number of genes.

Given the extracted representation vectors for each element in the input gene expression vector, we feed them into self-attention layers[13] to learn enhanced representations of these elements. Self-attention computes pairwise correlations between elements, capturing the relationships among them. To balance computational efficiency with representation effectiveness, we employ a large Performer[25] encoder and a mini Performer encoder to process elements with varying expression magnitudes. First, we rank the elements in the gene expression vector in descending order of expression values

and select the top 4096 with the highest values for processing by the large Performer encoder. This encoder applies self-attention across all 4096 high-expression elements, comprising 42 Performer layers with 32 attention heads and a hidden dimension of 1,280, and produces 200-dimensional output vectors. The remaining $K - 4,096$ elements, where $K = 27,874$ represents the total number of genes, are processed by a mini Performer encoder tailored for lower expression values. This encoder performs self-attention across all $K - 4,096$ elements. This mini encoder has 2 layers, 8 attention heads, and a hidden dimension of 200, yielding 200-dimensional output representations as well. After processing by these encoders, each element in the input expression vector has a 200-dimensional representation, derived from either the large or mini encoder. These representations are then fed into a full-length Performer encoder, which performs self-attention across all 27,874 elements. This final encoder has 2 layers, 8 attention heads, and a hidden dimension of 200. The resulting representations from the full-length encoder serve as the final outputs of scLong and are utilized for a range of downstream tasks.

## scLong pretraining

scLong was pretrained using a masked value reconstruction task. In this approach, 15% of the non-zero values in each input gene expression vector were randomly masked, and the model was trained to predict the masked values based on the unmasked portions of the vector. The 15% masking ratio followed that used in BERT[12]. Let $M_x$ represent the set of indices corresponding to masked gene expressions in an input gene expression vector $\mathbf{x}$. We create a masked expression vector $\mathbf{x}'$ by assigning a special symbol [MASK] to each masked gene while leaving unmasked values intact:

$$x'_i = \begin{cases} [\text{MASK}], & \text{if } i \in M_x \\ x_i, & \text{if } i \notin M_x \end{cases}$$

We then obtain a representation vector for each element in $\mathbf{x}'$. For unmasked expression values $x_i$, we apply an MLP to generate their representation vectors as previously described. For each [MASK] symbol, we use a learnable representation vector specific to [MASK]. These representation vectors for elements in $\mathbf{x}'$ are subsequently fed into the remaining layers of scLong to compute a final representation for each element. Finally, the representation vector corresponding to each masked gene is processed through a gene-specific MLP, producing a scalar representing the reconstructed value for that gene's masked expression. Let $\hat{x}_i$ and $x_i$ represent the reconstructed value and the ground truth (pre-masking) value of a masked gene $i$, respectively. The reconstruction loss is measured as the MSE between $\hat{x}_i$ and $x_i$. Pretraining is performed by minimizing the reconstruction loss across the dataset. Let $D$ denote the entire pretraining dataset. The overall pretraining loss is defined as:

$$\mathcal{L} = \frac{1}{|D||M_x|} \sum_{x \in D} \sum_{i \in M_x} (\hat{x}_i - x_i)^2, \tag{2}$$

where $| \cdot |$ denotes the size of the set.

During pretraining, we divided the 48 million cells into two sets: 95% for training and 5% for validation. The pretraining process was implemented using PyTorch Distributed Data Parallelism[75] and half-precision BFLOAT16 operations. The model was trained across 12 machines, each equipped with 8 A100 GPUs (80GB memory per GPU). Training was conducted over 5 epochs, taking a total of 35 GPU days. The batch size per GPU was set to 1, with a gradient accumulation step size of 200. We employed the Adam[76] optimizer with default hyper-parameters: $\beta_1 = 0.9$, $\beta_2 = 0.999$, $\epsilon = 10^{-8}$, and no weight decay. A cosine annealing scheduler was used to adjust the learning rate. The first cycle step size was 15, with cycle step magnification of 2. The maximum and minimum learning rates for the first cycle were $5 \times 10^{-5}$ and $10^{-6}$,

respectively. The linear warmup step size was 5. The max learning rate decreased by a factor of 0.9 in each cycle.

## Prediction of transcriptional responses to genetic perturbations

For this task, we used the Norman dataset[27], preprocessed with GEARS[14], which includes 91,205 cell samples and 5,045 genes. The dataset features 236 perturbation conditions: 105 involving single-gene perturbations, such as "FOXA1" and "HOXB9", and 131 involving double-gene perturbations, such as "{ZBTB10, SNAI1}", "{CDKN1A, CDKN1B}", and "{FOXA1, HOXB9}". Each double-gene perturbation is a combination of two single-gene perturbations. We adopted the data split used in GEARS, comprising a training set of 58,134 cells, a validation set of 6792 cells, and a test set of 26,279 cells. Each test sample was assigned to one of four categories: (1) neither gene in a double-gene perturbation appears in the training data (Seen 0/2); (2) one gene in a double-gene perturbation is absent from the training data (Seen 1/2); (3) both genes in a double-gene perturbation are present in the training data (Seen 2/2); and (4) the gene in a single-gene perturbation is absent from the training data (Seen 0/1).

Supplementary Fig. 4 illustrates the model architecture used for this downstream task. The input includes a gene expression vector of a cell prior to perturbation and the associated perturbation condition. The output is the gene expression vector of the cell following perturbation. We use the pretrained scLong model to derive a representation vector for each element of the pre-perturbation expression vector, while the GEARS method generates a representation for the perturbation condition. These vectors are then combined and processed through the GEARS decoder to predict the post-perturbation gene expression vector. Specifically, GEARS generates a 200-dimensional representation vector for each single-gene perturbation. For a double-gene perturbation, its representation is obtained by summing the vectors of the two individual single-gene perturbations it comprises. The representation vector for the perturbation condition is added to the representation vector of each element in the gene expression vector extracted by scLong. A ReLU activation is then applied to each dimension of the resulting vectors. Each vector is subsequently passed through a three-layer MLP with hidden dimensions of 200, 400, and 200, followed by a batch normalization[77] layer, producing a 200-dimensional post-perturbation representation for each expression element. Finally, each post-perturbation representation vector is processed by a decoder to generate post-perturbation values. The decoder begins with a one-layer MLP, which takes a post-perturbation representation as input and outputs an initial predicted post-perturbation value. Simultaneously, the decoder concatenates the post-perturbation representations of all expression elements, passing this combined vector through a two-layer MLP (with hidden dimensions of 5045 and 200) to produce a 200-dimensional vector. This vector is then concatenated with the initial predicted post-perturbation value of each element and fed into another MLP, which outputs an additional scalar prediction for each element. This scalar is then added to the corresponding pre-perturbation expression value to yield the final predicted post-perturbation value.

We assess the discrepancy between predicted and ground-truth post-perturbation expression vectors, denoted as $\hat{\mathbf{y}}$ and $\mathbf{y}$ respectively, using a composite loss function adopted from GEARS, which includes an autofocus loss and a direction-aware loss. Let $c$ denote the perturbation condition corresponding to $\mathbf{y}$ and $S_c$ represent the set of post-perturbation expression vectors in the training data resulting from applying $c$. Define $Z_c$ as the subset of genes that exhibit non-zero expression in at least one vector within $S_c$. The autofocus loss is defined as follows:

$$\mathcal{L}_{\text{af}} = \frac{1}{|Z_c|} \sum_{k \in Z_c} (\hat{y}_k - y_k)^4. \tag{3}$$

Let $\mathbf{x}$ denote the pre-perturbation expression vector corresponding to $\mathbf{y}$. The direction-aware loss function assesses the alignment of directional changes between the predicted and actual post-perturbation expressions relative to their pre-perturbation states:

$$\mathcal{L}_{\mathrm{d}} = \frac{1}{|Z_c|} \sum_{k \in Z_c} \left[ \mathrm{sign}(\widehat{y}_k - x_k) - \mathrm{sign}(y_k - x_k) \right]^2, \qquad (4)$$

where $\mathrm{sign}(\cdot)$ denotes the sign function, which determines the sign of a real number $a$:

$$\mathrm{sign}(a) = \begin{cases} -1 & \text{if } a < 0, \\ 0 & \text{if } a = 0, \\ 1 & \text{if } a > 0. \end{cases}$$

The model's total loss function is a sum of the autofocus loss and the direction-aware loss. During training, the model aims to minimize this total loss across all data examples.

The hyperparameters for our method were mostly the same as those used in GEARS. The hidden dimension of the GEARS encoder and decoder was set to 1024, with ReLU as the activation function. Model weights were optimized using the Adam[76] optimizer ($\beta_1 = 0.9$, $\beta_2 = 0.999$, $\epsilon = 10^{-8}$, $\lambda = 5 \times 10^{-4}$, learning rate $\gamma = 10^{-3}$). Training was conducted across 4 GPUs, with a batch size of 16 per GPU. With 8 gradient accumulation steps, the effective overall batch size was $16 \times 8 \times 4 = 512$. Our model was trained for 16 epochs, while GEARS was trained for 20 epochs to replicate their reported results. For baseline foundation models with available fine-tuning implementations for this downstream task, such as scGPT and scFoundation, we directly used their provided code. For those without such implementations, we fine-tuned them using the same approach as for scLong. For each method, early stopping was applied when performance on the validation set started to decline.

To evaluate the model's performance, we employed two metrics: MSE and PCC, focusing on the top 20 DE genes. The set of top 20 DE genes for a given perturbation condition $c$, denoted as $D_c$, was identified as the 20 genes with the highest variance across the expression vectors in $S_c$, the set of post-perturbation expression vectors under condition $c$. For each expression vector —predicted ($\widehat{\mathbf{y}}$), ground truth ($\mathbf{y}$), and pre-perturbation ($\mathbf{x}$)—under perturbation condition $c$, we extracted the subvector corresponding to $D_c$, denoted as $\widehat{\mathbf{y}}[D_c]$, $\mathbf{y}[D_c]$, and $\mathbf{x}[D_c]$, respectively. The MSE on $D_c$ is calculated as:

$$\mathrm{MSE}(D_c) = \frac{1}{|D_c|} \sum_{k \in D_c} (\widehat{y}_k - y_k)^2. \qquad (5)$$

The PCC on $D_c$ assesses the correlation between the predicted and actual changes in expression from pre- to post-perturbation:

$$\mathrm{PCC}(D_c) = \mathrm{PCC}(\widehat{\mathbf{y}}[D_c] - \mathbf{x}[D_c], \mathbf{y}[D_c] - \mathbf{x}[D_c]), \qquad (6)$$

where the Pearson Correlation Coefficient $\mathrm{PCC}(\mathbf{v}, \mathbf{u})$ between two $n$-dimensional vectors, $\mathbf{v}$ and $\mathbf{u}$, is given by:

$$\mathrm{PCC}(\mathbf{v}, \mathbf{u}) = \frac{\sum_{i=1}^{n} (v_i - \bar{v})(u_i - \bar{u})}{\sqrt{\sum_{i=1}^{n}(v_i - \bar{v})^2} \sqrt{\sum_{i=1}^{n}(u_i - \bar{u})^2}}. \qquad (7)$$

In this formula, $\bar{v}$ and $\bar{u}$ represent the mean values of $\mathbf{v}$ and $\mathbf{u}$, respectively. For each of the four test sample categories—Seen 0/2, Seen 1/2, Seen 2/2, and Seen 0/1—we calculated the $\mathrm{MSE}(D_c)$ and $\mathrm{PCC}(D_c)$ for each sample. The overall performance for each category was then determined by averaging $\mathrm{MSE}(D_c)$ and $\mathrm{PCC}(D_c)$ across all samples in that category.

In classifying gene interaction types, we utilize a magnitude score[14]. For a double-gene perturbation $\{i, j\}$, scLong's magnitude score is defined as follows. Let $\mathbf{x}$ denote a cell's pre-perturbation expression vector and $\widehat{\mathbf{y}}$ represent the post-perturbation expression vector predicted by scLong for the same cell under perturbation condition $\{i, j\}$. The difference $\widehat{\mathbf{y}} - \mathbf{x}$ is considered the perturbation effect of $\{i, j\}$ for this cell. Repeating this process for all test cells, we calculate the average perturbation effect, denoted as $\mathbf{\Delta}_{i,j}$. Similarly, we compute the average perturbation effect $\mathbf{\Delta}_i$ for single-gene perturbation $i$ and $\mathbf{\Delta}_j$ for single-gene perturbation $j$. We then solve the following equation:

$$\mathbf{\Delta}_{i,j} = \alpha \mathbf{\Delta}_i + \beta \mathbf{\Delta}_j + \mathbf{c}, \qquad (8)$$

where $\alpha$ and $\beta$ are scalars representing the linear combination coefficients, and $\mathbf{c}$ is an offset vector. The magnitude score is then defined as $\sqrt{\alpha^2 + \beta^2}$. Hierarchical clustering in Fig. 2e was conducted using the Seaborn library[78].

## Prediction of transcriptional responses to chemical perturbations

In this task, we used a subset of the L1000 dataset[32], which comprises 7 distinct cell lines, 978 genes, and 810 drug compounds, each tested at 6 dosage levels. The prediction model takes two inputs: (1) the index of a perturbed cell line, and (2) the molecular graph and dosage level of the drug used to induce the perturbation. The model's output is the gene expression profile of the cell line following perturbation. The dataset does not provide pre-perturbation gene expression data. Each example in the L1000 dataset includes these inputs and outputs, with a total of 5005 examples divided into 3965 for training, 544 for validation, and 496 for testing.

Supplementary Fig. 5 illustrates the model architecture for this task. The perturbed cell line is encoded using a 7-dimensional one-hot vector, where each dimension represents one of the 7 cell lines, and each is associated with a 4-dimensional learnable embedding. Similarly, the drug dosage is encoded using a 6-dimensional one-hot vector, with each dimension corresponding to one of the 6 dosages, and each is linked to a 4-dimensional learnable embedding. For each of the 978 genes, scLong extracts a 200-dimensional representation vector as the output of its GCN built on the gene graph. This vector is then passed through a linear projection layer to generate a 512-dimensional representation. We employ a GCN to extract a representation vector for each atom in the input drug molecule graph. The GCN consists of three convolutional layers, each with a hidden dimension of 128. Next, cross-attention[13] is applied between genes and drug atoms to capture their interaction patterns. Specifically, the representation vectors of drug atoms extracted by the GCN serve as both the key and value vectors in the cross-attention module, while the gene representation vectors, obtained from scLong and the subsequent linear layer, serve as the query vectors. Prior to cross-attention, the drug atom representations are mapped to 512-dimensional vectors via a linear projection layer to align with the gene representation dimensions. The cross-attention module consists of 2 attention layers, 4 attention heads, and a hidden dimension of 512. Finally, the gene representations from the cross-attention module are integrated with drug, cell line, and dosage information. Specifically, the 512-dimensional representation vector of each gene obtained from cross-attention is concatenated with a 4-dimensional cell line embedding, a 4-dimensional dosage embedding, and a 128-dimensional drug representation vector averaged across the representations of all atoms in the drug. The resulting concatenated vector is then passed through a 2-layer MLP to predict the post-perturbation expression for each gene. These two layers have dimensions of 648 and 1, respectively, with ReLU serving as the activation function. The predictions for each gene are concatenated to form the final predicted gene expression vector.

The model was trained by minimizing the MSE between the predicted and ground-truth post-perturbation gene expression vectors, using the Adam optimizer with a learning rate of 2e-4, betas of (0.9, 0.999), epsilon of 1e-8, and no weight decay. A linear learning rate scheduler was used, with 5 warmup epochs and a final learning rate of 2e-5. The training was conducted with a batch size of 16 for a maximum of 100 epochs. The final performance was evaluated on the test set, using the checkpoint that achieved the best validation performance.

We compared our method with Geneformer, scGPT, scFoundation, UCE, and the task-specific DeepCE model, all of which have similar model configurations, differing only in their approaches for gene representation extraction. In DeepCE, gene representations are derived from the STRING protein interaction network[59] using the Node2Vec method[79]. Baseline foundation models extract representations using their pretrained models. Spearman and Pearson correlation coefficients, root mean squared error (RMSE), and precision for top-$K$ positive and negative predictions were used as evaluation metrics. Given predicted and ground truth gene expression vectors $\mathbf{x} = (x_1, ..., x_n)$ and $\mathbf{y} = (y_1, ..., y_n)$, we first rank the values in each vector in descending order. Let $R(\mathbf{x}) = (R(x_1), ..., R(x_n))$ and $R(\mathbf{y}) = (R(y_1), ..., R(y_n))$ represent the rank positions for values in $\mathbf{x}$ and $\mathbf{y}$, respectively. The Spearman correlation (SC) score is defined as: $SC(\mathbf{x}, \mathbf{y}) = \rho(R(\mathbf{x}), R(\mathbf{y}))$, where $\rho$ denotes Pearson correlation. The root mean squared error (RMSE) between $\mathbf{x}$ and $\mathbf{y}$ is calculated as:

$$RMSE(\mathbf{x}, \mathbf{y}) = \sqrt{\frac{1}{n}\sum_{i=1}^{n}(x_i - y_i)^2}. \tag{9}$$

To compute the positive precision at $K$ (Pos-P@K), we identify the top-$K$ genes with the highest expression values in $\mathbf{y}$ as $G_y$ and in $\mathbf{x}$ as $G_x$. Pos-P@K is defined as:

$$Pos - P@K = \frac{1}{K}|G_y \cap G_x|, \tag{10}$$

reflecting the proportion of genes in the predicted set $G_x$ that are also present in the ground truth set $G_y$. The negative precision at $K$ (Neg-P@K) is computed analogously, using the lowest-expressed genes in $\mathbf{x}$ and $\mathbf{y}$. In our calculations, we set $K = 100$.

### Prediction of cancer drug responses

The dataset[36] for this task was created by combining the Cancer Cell Line Encyclopedia[80] and the Genomics of Cancer Drug Sensitivity[81] datasets, resulting in 561 cancer cell lines and 238 drugs. Each cell line is represented by a bulk gene expression vector of 697 genes. Out of the total 561 × 238 = 133,518 possible (cell line, drug) interaction pairs, ~19.5% (26,072) had missing IC50 values, leaving 107,446 complete pairs. From these, 95% were allocated for training and 5% for testing.

Supplementary Fig. 6 illustrates the model architecture for this task. The input consists of the molecular structure of a potential cancer drug and the bulk gene expression profile of a cancer cell line. The output is a prediction of the drug's efficacy against the cancer cell line, quantified by its half-maximal inhibitory concentration (IC50) value. We use the scLong model to extract a representation vector from the gene expression data, which is concatenated with the drug molecule representation obtained via a GCN. This combined representation is then passed through a regression module to predict the IC50 value. Specifically, the scLong model processes a 697-dimensional gene expression vector as input, generating a 100-dimensional representation for each gene, resulting in a 697 × 200 matrix. We apply average pooling across the 200 dimensions to reduce this matrix to a 697-dimensional vector. This representation is then passed through a two-layer MLP, with hidden dimensions of 256 and 100, respectively. The output of this MLP is a final 100-dimensional representation that captures the cell line's features. The GCN consists of 3 convolutional layers, each with a hidden dimension of 100 and using the ReLU activation function. It learns a 100-dimensional representation for each atom in the molecular graph. To obtain a single representation for the entire molecule, average pooling is applied across the atom-level vectors. The regression module comprises a one-layer MLP with a hidden dimension of 300, followed sequentially by a convolutional neural network (CNN) with 3 layers and a final linear layer that outputs a scalar. The CNN layers have filter numbers and kernel sizes of (30, 150), (10, 5), and (5, 5), respectively. The scalar from the linear layer is passed through a sigmoid function to predict the IC50 value. To prevent overfitting, a dropout[82] rate of 0.1 is applied to all layers.

The model was trained using a MSE loss function, which measures the difference between the predicted and ground truth IC50 values. We optimized the model parameters using the Adam optimizer[76], with $\beta_1 = 0.9$, $\beta_2 = 0.999$, and $\epsilon = 10^{-8}$. The learning rate was set to a fixed value of 0.001, without the use of a learning rate scheduler. Training was performed for up to 500 epochs with a batch size of 64, using early stopping based on validation loss.

We compared scLong with other foundation models, including Geneformer, scGPT, scFoundation, and UCE. We also evaluated its performance against task-specific models, such as the deep neural network DeepCDR[36] and a linear model[36,37]. The only difference between our method and DeepCDR as well as the baseline foundation models lies in how the representation vectors are extracted from the input gene expression data; the rest of the model architecture and hyperparameter settings remain identical. In DeepCDR, the raw gene expression vector is directly fed into the regression module without learning additional representations. For foundation models, we used the pretrained model to extract a representation from the gene expression vector before passing it to the regression module. We used Pearson correlation, defined similarly to Equation (7), as the evaluation metric.

### Prediction of cancer cell line responses to synergistic drug combinations

In this task, each data sample consists of a bulk gene expression vector from a cancer cell line, a pair of drugs, and a binary label indicating whether the drug combination is effective against the cell line. The training dataset, sourced from[39], includes 12,415 examples, spanning 36 anti-cancer drugs and 31 human cancer cell lines. The test dataset, obtained from an independent source[40], contains 668 examples, covering 57 drugs and 24 cancer cell lines. Both datasets include gene expression values for 954 genes.

Supplementary Fig. 7 illustrates the model architecture for this task, which closely resembles the architecture used for single-drug response prediction (Supplementary Fig. 6). First, the 954-dimensional gene expression vector is processed by the scLong model, yielding a (954, 200) representation matrix. After applying average pooling across the 200 dimensions, we obtain a 954-dimensional representation vector. This vector is then passed through a 3-layer MLP with hidden dimensions of 512, 256, and 128, using the ReLU activation function. For the drug pair input, two separate 3-layer GCNs are employed to generate a representation for each drug. The GCN layers use ReLU as the activation function, with hidden dimensions of 1024, 512, and 156, respectively. The representations of the two drugs are concatenated with the gene expression representation obtained from the MLP. The combined representation is then passed through another 3-layer MLP to predict the binary output label, with ReLU activation and hidden dimensions of 1024, 512, and 128, respectively.

The model was optimized using cross-entropy loss with the Adam optimizer[76]. The learning rate was set to 1e-4, with $\beta_1 = 0.9$, $\beta_2 = 0.999$, $\epsilon = 1e - 08$, and no weight decay. We used a linear learning rate scheduler with 50 warmup epochs and a final learning rate of 1e-5. The model was trained with a batch size of 256 for 1000 epochs. The

baseline method, DeepDDS[41], directly used raw gene expression vectors as the representations of cell lines, without learning additional latent representations. For the baseline foundation models, including Geneformer, scGPT, scFoundation, and UCE, the pretrained models were used to extract representations from the gene expression vectors. All other model settings and hyperparameters were kept consistent with our method. The performance of the models in this binary classification task was evaluated using the AUROC score.

## Inference of gene regulatory networks

The input for this task consists of gene expression vectors from a collection of cells, with the output being a GRN represented as an adjacency matrix. In this matrix, each element indicates the interaction strength between two genes. We utilized gene expression data from $N_c = 758$ hESC[44], covering $N_g = 17, 735$ genes.

Supplementary Fig. 8 presents the model architecture used for this task. Initially, we applied the pretrained scLong model to extract a $N_g \times 200$ representation matrix from each cell's gene expression vector, where each row is a 200-dimensional representation vector of gene expression elements. Aggregating these matrices across cells yields a $N_c \times N_g \times 200$ tensor. By averaging over the last dimension of this tensor, we obtain a $N_c \times N_g$ matrix, where each column serves as a new representation vector for a gene across all cells. We then compute a $N_g \times N_g$ adjacency matrix **A** by calculating the cosine similarity between the $N_c$-dimensional representation vectors of each pair of genes, capturing gene-gene relationships. Following DeepSEM[15], we use the preliminary adjacency matrix **A** as an initialization for further refinement with a beta variational autoencoder (beta-VAE)[46]. The beta-VAE framework uses **A** to model gene expression vectors and consists of a probabilistic encoder and decoder. The encoder defines a conditional distribution $p(\mathbf{z}|\mathbf{x})$, where **x** is a gene expression vector and **z** is a 128-dimensional latent vector representing **x**. The decoder defines a conditional distribution $p(\mathbf{z}|\mathbf{x})$. Both distributions are parameterized as multivariate Gaussians, with the mean and covariance for $p(\mathbf{z}|\mathbf{x})$ computed by an encoder network receiving **x** as input, and for $p(\mathbf{z}|\mathbf{x})$, by a decoder network taking **z** as input. The encoder network comprises a three-layer MLP followed by a GRN layer parameterized by **A**, while the decoder network includes a reverse GRN layer also parameterized by **A**, followed by a three-layer MLP. The GRN layer applies a linear transformation with parameters $\mathbf{I} - \mathbf{A}$, and the reverse GRN layer applies another linear transformation with parameters $(\mathbf{I}-\mathbf{A})^{-1}$. The total loss for the beta-VAE, a weighted sum of reconstruction and KL divergence losses, optimizes the encoder and decoder MLPs and refines the adjacency matrix **A** to produce the final inferred GRN, leveraging the reparameterization trick[83]. Both the encoder and decoder MLPs have a hidden dimension of 128 and use Tanh as the activation function.

The model was optimized using the RMSProp optimizer[84] with a learning rate of 2e-5 for the adjacency matrix and 1e-4 for other parameters. We set $\alpha = 0.99$, $\epsilon = 1e − 08$, weight decay to 0, and momentum to 0. A linear learning rate scheduler was applied with a step size of 0.99 and $\gamma = 0.95$. Training was conducted with a batch size of 64 over 120 epochs.

We evaluated scLong against several foundation models, including Geneformer, scGPT, scFoundation, UCE, as well as task-specific approaches such as DeepSEM and GENIE3[48]. For comparison, we also included a straightforward baseline, *GO Graph*, which constructs the GRN for the 17,735 genes by directly using the corresponding subgraph of the gene graph (Fig. 1a) derived from the GO. The primary difference between DeepSEM and foundation models lies in how each approach initializes the adjacency matrix **A**. DeepSEM initializes **A** with a uniform distribution in the range (0, 2e-4). Foundation models initialize **A** using cosine similarity between expression representations derived from their pretrained models. To evaluate these methods, we used a GRN derived by ChIP-Seq[85] as the ground truth. This network comprises 2,762 nodes representing genes, 436,563 edges representing gene interactions, and includes 487 transcription factors (TFs). We compared the GRNs inferred by different methods against this ground-truth GRN. Given the high sparsity of the ground-truth network—only a small fraction of gene pairs exhibit regulatory interactions (436,563 out of a possible $17,735 \times 17,735$ pairs)—we employed EPR[44] and AUPR[44] as evaluation metrics. EPR is defined as the ratio between the Pos-P@K score of an inferred adjacency matrix (as previously specified) and that of a random predictor. Here, we set $K$ to 436, 563, which corresponds to the number of edges in the ground truth GRN. Specifically, the edges with the top-K values from the inferred adjacency matrix **A** form an edge set $E_p$, and we calculate the proportion of these edges that also appear in the ground truth edge set $E_g$ using the formula $\frac{1}{K}|E_p \cap E_g|$. The random predictor estimates the presence of an edge between a gene pair with a probability of $p = 436, 563/(17, 735 \times 17, 735)$, which represents the ratio of edges in the ground truth GRN to the total possible edges. The Pos-P@K score for the random predictor is thus $p$. AUPR is computed as the ratio of the area under the precision-recall curve (AUPRC) for the inferred adjacency matrix to that of the random predictor, where the random predictor's AUPRC equals $p$.

## Zero-shot batch integration

For the zero-shot batch integration task, we used the widely adopted pancreas dataset[49], which comprises $M = 16, 382$ cells and $N = 19, 093$ genes across six batches, each generated using a distinct scRNA-seq technology. The gene expression vector $\mathbf{x} \in \mathbb{R}^N$ of each cell is processed by the pretrained scLong model to produce a cell-level embedding. Let $\mathbf{E} \in \mathbb{R}^{N \times D}$ denote the contextualized representations produced by the full-length Performer encoder (Fig. 1a) for the $N$ genes, where $D = 200$ is the embedding dimension for each gene. Let $\mathbf{x}' \in \mathbb{R}^N$ denote the reconstructed gene expression vector generated by the model's MLP decoder (Fig. 1b). We concatenate **E** and $\mathbf{x}'$ to form a matrix $\mathbf{V} \in \mathbb{R}^{N \times (D+1)}$. We then apply sum pooling across the rows of **V** to obtain the final cell representation $\mathbf{v} \in \mathbb{R}^N$:

$$v_i = \sum_{j=1}^{D+1} V_{ij}. \tag{11}$$

## Ablation studies of scLong

We conducted ablation studies on downstream tasks to evaluate the impact of two key components of scLong: (1) modeling LEGs and (2) integrating the GO graph. To evaluate the contribution of LEGs, we compared the full scLong model—which incorporates both high-expression genes and LEGs—with two ablated variants. In the first variant, *w/o LEG*, we excluded LEGs entirely, retaining only the top 4096 high-expression genes per cell. Correspondingly, the mini Performer encoder responsible for processing LEGs was removed. The output of the large Performer encoder, which processes the 4096 high-expression genes, was fed directly into the subsequent full-length Performer encoder. In the second variant, *Random LEG*, we preserved the mini Performer encoder's architecture (2 layers, 8 attention heads, hidden dimension of 200) but replaced its learned weights with values drawn independently from a normal distribution, i.e., each parameter was sampled from $\mathcal{N}(0, 1)$.

For the second study evaluating the impact of the GO graph, we compared the full scLong model against two ablated variants. In the *w/o GO* setting, we removed the GO graph and its associated GCN, using only Gene2Vec embeddings as the final gene representations, which were directly added to the expression representations. In the *Random GO* setting, we replaced the GO graph with a randomly generated graph. Specifically, each edge weight in the GO graph was randomized by sampling from a uniform distribution, i.e., each edge weight was sampled from $\mathcal{U}(0, 1)$.

These studies were conducted across three tasks: predicting transcriptional outcomes of genetic perturbations, inferring GRNs, and zero-shot batch integration. For each task, the experimental setup —including model hyperparameters, optimizer, learning rate scheduler, and so on—was kept consistent with the original scLong configuration.

## Clustering of gene representations extracted by scLong

For each cell type in the Zheng68K dataset[55], which includes 11 cell types, 65,000 cells and 20,000 genes, we randomly sampled 50 cells. Using the pretrained scLong model, we extracted representations for each cell's gene expression elements. To generate an overall representation vector for each gene, we averaged its representations across the 50 sampled cells. We then calculated the cosine similarity between each gene pair, where the cosine similarity of two vectors, $\mathbf{x}$ and $\mathbf{y}$, is given by $\frac{\mathbf{x} \cdot \mathbf{y}}{\|\mathbf{x}\| \|\mathbf{y}\|}$, with $\mathbf{x} \cdot \mathbf{y} = \sum_i x_i y_i$ as the dot product and $\|\mathbf{x}\| = \sqrt{\mathbf{x} \cdot \mathbf{x}}$ as the $L^2$ norm. Cosine similarity values range from -1 to 1. We selected the 50 genes with the highest cumulative similarity scores for further analysis. Hierarchical clustering was then performed on the similarity matrix using the *clustermap* function from the Seaborn[78] library. The Zheng68K dataset includes known marker genes identified from prior studies for each cell type[55]. Marker genes, or cell-type-specific genes, are typically expressed at high levels in a specific cell type and at low levels in others[86]. These genes are essential for manual or semi-supervised cell classification[87,88], and the dataset provider used them to classify cells into the 11 defined types.

## Reporting summary

Further information on research design is available in the Nature Portfolio Reporting Summary linked to this article.

## Data availability

The pretraining datasets were collected from public datasets hosted on CELLxGENE (https://cellxgene.cziscience.com/datasets), Cell Blast (https://cblast.gao-lab.org/), and the Human Cell Atlas (https://www.humancellatlas.org/). The datasets used for downstream tasks are accessible from the following links: genetic perturbation dataset (https://github.com/snap-stanford/GEARS); chemical perturbation dataset (https://github.com/njpipeorgan/L1000-bayesian); single drug and drug combination response datasets (https://github.com/kimmo1019/DeepCDR and https://github.com/Sinwang404/DeepDDS); GRN inference datasets (https://github.com/HantaoShu/DeepSEM); zero-shot batch integration dataset (https://figshare.com/articles/dataset/Benchmarking_atlas-level_data_integration_in_single-cell_genomics_-_integration_task_datasets_Immune_and_pancreas_/12420968); and marker gene clustering dataset (https://zenodo.org/records/3357167). The datasets curated and utilized in this study, trained model parameters, and other files necessary to reproduce the experimental results, figures, and tables can be accessed at https://mbzuaiac-my.sharepoint.com/:f:/g/personal/ding_bai_mbzuai_ac_ae/EpvKzQW4hI5Bnb88-iM7vE0B_e2_U5r_ZGXb_FILCLTw3Qand https://figshare.com/account/articles/30105148. Source data are provided with this paper.

## Code availability

The source code for this work is available at https://github.com/BaiDing1234/scLongand is archived at https://zenodo.org/records/17510567[89].

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

## Acknowledgements
P.X. acknowledges funding support from NIH R35GM157217, NSF IIS2405974, and NSF IIS2339216. E.X. acknowledges funding support from NSF CNS2414087, NSF BCS2040381, NSF IIS2123952, NSF IIS1955532, NSF IIS2311990, NIH R01GM140467, NGA HM04762010002, SRC AIHW award 2024AH3210, NIGMS R01GM140467, and DARPA ECOLE HR00112390063.

## Author contributions
D.B., S.M., and R.Z. contributed to conceptualization, methodology, software, investigation, analysis, writing-original draft, and writing-editing. Y.L. contributed to conceptualization, methodology, and software. J.G., J.Y., Q.W., H.R., T.A., D.G., S.Z., N.L., W.W., and T.I. contributed to investigation, analysis, and writing-editing. P.X. and E.X. contributed to conceptualization, methodology, investigation, analysis, writing-original draft, and writing-editing.

## Competing interests
T.I. is a cofounder, member of the advisory board and has an equity interest in Data4Cure and Serinus Biosciences. T.I. is a consultant for and has an equity interest in IDEAYA Biosciences. The terms of these arrangements for T.I. have been reviewed and approved by the University of California, San Diego, in accordance with its conflict of interest policies. The remaining authors declare no competing interests.
