## [Transparent Peer Review file · Nature Communications]

scLong: A Billion-Parameter Foundation Model for Capturing Long-Range Gene Context in Single-Cell Transcriptomics

Corresponding Author: Professor Pengtao Xie

Version 0:

Reviewer comments:

Reviewer #1

(Remarks to the Author)

We would like to commend the authors for addressing our comments. The results seem convincing.

Our only final comment is in terms of statistical tests - we think the authors should perform some type of multiple hypothesis correction, at least at the level of the individual benchmarks.

(Remarks on code availability)

The code has improved and is now a more proper pipeline.

Version 1:

Reviewer comments:

Reviewer #2

(Remarks to the Author)

The authors have addressed all my comments in a thoughtful manner. Congratulations on the improvements and the quality of the final manuscript.

(Remarks on code availability)

Response Letter for “scLong: A Billion-Parameter Foundation Model for Capturing Long-Range Gene Context in Single-Cell Transcriptomics”

Ding Bai *, Shentong Mo *, Ruiyi Zhang *, Yingtao Luo, Jiahao Gao, Jeremy Parker Yang, Qiuyang Wu, Digvijay Singh, Hamidreza Rahmani, Tiffany Amariuta, Danielle Grotjahn, Sheng Zhong, Nathan Lewis, Wei Wang, Trey Ideker, Pengtao Xie ✉, Eric Xing ✉

We sincerely thank the reviewers and the editor for their thoughtful and constructive feedback in this round of review. We have carefully revised the manuscript to address the remaining point regarding multiple hypothesis testing. Specifically, we applied the Benjamini–Hochberg procedure to adjust for multiple comparisons and updated the main manuscript and supplementary materials. We are confident that the revised version fully addresses the comments of the reviewers and the editor and further improves the manuscript.

*Equal contributions. ✉Correspondence: p1xie@ucsd.edu and epxing@cs.cmu.edu

Contents

1	Responses to Reviewer #1	3
1.1	Multiple hypothesis correction of P-values at the level of each benchmark	3
1.2	Improved code availability	3

1 Responses to Reviewer #1

1.1 Multiple hypothesis correction of P-values at the level of each benchmark

Reviewer #1 (Remarks to the Author):

We would like to commend the authors for addressing our comments. The results seem convincing.

We thank the reviewer for the positive feedback and are glad that the results are convincing.

Our only final comment is in terms of statistical tests - we think the authors should perform some type of multiple hypothesis correction, at least at the level of the individual benchmarks.

We thank the reviewer for the valuable suggestion regarding multiple hypothesis correction. In response, we have applied the Benjamini–Hochberg procedure [1] to adjust the results of the pairwise t -tests at the level of the individual benchmarks, as recommended, and updated the manuscript accordingly. The majority of results remain significant after correction, as shown in Tables 1, 2, 3, 4, 5 and 6.

These adjusted P-values have been incorporated into the Results section of the revised manuscript. The t -test results are in Section 6 and Table 11-16 of the Supplementary Material.

Response Table 1: **Two-sided t -test statistics comparing scLong with other baseline models in predicting transcriptional outcomes of genetic perturbations.** All models, including scLong, were evaluated across five independent runs (sample size = 5). Adjusted P values were calculated using the Benjamini–Hochberg procedure for multiple hypothesis correction [1, 3]. Effect sizes are reported using Cohen’s d [2].

Scenario	Baseline	MSE			Pearson		
		P	Adjusted P	Effect size	P	Adjusted P	Effect size
Seen 0/2	GEARS	< 0.001	0.001	5.991	0.003	0.003	4.176
	Geneformer	0.107	0.107	1.305	< 0.001	0.001	7.171
	scGPT	0.004	0.005	3.720	0.162	0.162	1.081
	scFoundation	0.003	0.005	3.977	0.001	0.002	5.327
	UCE	< 0.001	< 0.001	10.446	0.002	0.003	4.420
Seen 1/2	GEARS	0.011	0.019	2.801	0.050	0.063	1.751
	Geneformer	0.006	0.016	3.272	< 0.001	0.002	5.809
	scGPT	0.024	0.030	2.226	0.084	0.084	1.441
	scFoundation	0.407	0.407	0.586	0.003	0.006	3.877
	UCE	< 0.001	< 0.001	16.907	< 0.001	0.002	6.369
Seen 2/2	GEARS	0.312	0.520	0.730	0.016	0.021	2.490
	Geneformer	0.001	0.003	5.362	0.007	0.012	3.178
	scGPT	0.649	0.649	0.309	0.401	0.401	0.593
	scFoundation	0.462	0.577	0.513	0.007	0.013	3.129
	UCE	< 0.001	< 0.001	9.707	< 0.001	0.001	7.948
Seen 0/1	GEARS	< 0.001	0.002	6.127	< 0.001	0.001	6.297
	Geneformer	0.383	0.383	0.619	0.001	0.002	5.062
	scGPT	0.253	0.316	0.843	0.002	0.002	4.552
	scFoundation	0.035	0.059	1.970	0.002	0.002	4.391
	UCE	< 0.001	0.002	5.877	< 0.001	< 0.001	15.581

1.2 Improved code availability

Reviewer #1 (Remarks on code availability):

The code has improved and is now a more proper pipeline.

We appreciate the reviewer’s recognition of our efforts to improve the code into a more proper pipeline.

References

- [1] Yoav Benjamini and Yosef Hochberg. Controlling the false discovery rate: a practical and powerful approach to multiple testing. *Journal of the Royal statistical society: series B (Method-*

Response Table 2: **Two-sided t -test statistics comparing scLong with other baseline models on the chemical perturbation task.** All models, including scLong, were evaluated across five independent runs (sample size = 5). Adjusted P values were calculated using the Benjamini–Hochberg procedure for multiple hypothesis correction [1, 3]. Effect sizes are reported using Cohen’s d [2].

Baseline	RMSE		Spearman		Pearson		Pos-P@100		Neg-P@100	
	Adjusted P	Effect size	Adjusted P	Effect size	Adjusted P	Effect size	Adjusted P	Effect size	Adjusted P	Effect size
DeepCE	0.031	2.064	0.006	3.864	0.003	4.776	0.007	3.162	0.033	2.056
Geneformer	0.002	5.547	0.014	2.649	0.008	3.018	0.006	3.616	0.033	2.014
scGPT	0.002	6.985	0.014	2.714	0.006	3.502	0.005	4.176	0.033	2.064
scFoundation	0.004	4.011	0.006	3.911	0.003	5.145	< 0.001	9.179	0.017	3.901
UCE	0.002	5.188	0.006	4.102	0.003	5.407	0.001	6.877	0.030	2.753

Response Table 3: **Two-sided t -test statistics comparing scLong with other baseline models on the single-drug and drug-combination response prediction tasks.** All models, including scLong, were evaluated across five independent runs (sample size = 5). Adjusted P values were calculated using the Benjamini–Hochberg procedure for multiple hypothesis correction [1, 3]. Effect sizes are reported using Cohen’s d [2].

Baseline	Pearson (single drug)			AUROC (drug combination)		
	P	Adjusted P	Effect size	P	Adjusted P	Effect size
DeepCDR	< 0.001	0.001	7.074	/	/	/
DeepDDS	/	/	/	< 0.001	0.001	6.499
Geneformer	0.001	0.001	5.210	0.006	0.006	3.376
scGPT	< 0.001	0.001	5.632	0.001	0.002	5.305
scFoundation	0.025	0.025	2.223	< 0.001	< 0.001	8.770
UCE	< 0.001	0.001	6.450	0.001	0.002	4.916

ological), 57(1):289–300, 1995.

- [2] Jacob Cohen. *Statistical power analysis for the behavioral sciences*. routledge, 2013.
- [3] Skipper Seabold and Josef Perktold. statsmodels: Econometric and statistical modeling with python. In *9th Python in Science Conference*, 2010.

Response Table 4: **Two-sided t -test statistics comparing scLong with other baseline models on the GRN inference task.** All models, including scLong, were evaluated across five independent runs (sample size = 5). Adjusted P values were calculated using the Benjamini–Hochberg procedure for multiple hypothesis correction [1, 3]. Effect sizes are reported using Cohen’s d [2].

Baseline	AUPR			EPR		
	P	Adjusted P	Effect size	P	Adjusted P	Effect size
GENIE3	< 0.001	< 0.001	27.969	< 0.001	0.001	6.982
DeepSEM	< 0.001	< 0.001	18.140	0.004	0.007	3.867
Geneformer	< 0.001	< 0.001	24.226	0.038	0.038	1.933
scGPT	< 0.001	< 0.001	17.846	0.010	0.014	2.942
scFoundation	< 0.001	< 0.001	35.938	< 0.001	< 0.001	18.128
UCE	0.001	0.001	5.008	0.016	0.019	2.515

Response Table 5: **Two-sided t -test statistics comparing scLong with the four ablation settings in predicting transcriptional outcomes of genetic perturbations.** All ablation settings, including scLong, were evaluated across five independent runs (sample size = 5). Adjusted P values were calculated using the Benjamini–Hochberg procedure for multiple hypothesis correction [1, 3]. Effect sizes are reported using Cohen’s d [2].

Scenario	Ablation setting	MSE			Pearson		
		P	Adjusted P	Effect size	P	Adjusted P	Effect size
Seen 0/2	w/o LEG	< 0.001	0.002	6.201	0.005	0.010	3.544
	Random LEG	0.009	0.009	2.958	0.041	0.041	1.879
	w/o GO	0.003	0.005	4.258	0.008	0.010	3.153
	Random GO	0.005	0.006	3.600	0.005	0.010	3.568
Seen 1/2	w/o LEG	0.003	0.005	3.923	0.056	0.056	1.688
	Random LEG	0.002	0.005	4.350	0.007	0.014	3.187
	w/o GO	0.134	0.134	1.188	0.002	0.007	4.758
	Random GO	< 0.001	0.003	5.857	0.020	0.027	2.354
Seen 2/2	w/o LEG	0.762	0.762	0.204	0.211	0.281	0.940
	Random LEG	0.094	0.188	1.383	0.009	0.038	2.946
	w/o GO	0.156	0.208	1.103	0.498	0.498	.470
	Random GO	0.0219	0.088	2.302	0.028	0.057	2.117
Seen 0/1	w/o LEG	0.075	0.075	1.509	0.107	0.142	1.309
	Random LEG	0.009	0.012	2.993	< 0.001	0.003	5.927
	w/o GO	0.003	0.007	4.229	0.645	0.645	0.314
	Random GO	0.003	0.007	3.923	0.005	0.010	3.511

Response Table 6: **Two-sided t -test statistics comparing scLong with the four ablation settings on the GRN inference task.** All ablation settings, including scLong, were evaluated across five independent runs (sample size = 5). Adjusted P values were calculated using the Benjamini–Hochberg procedure for multiple hypothesis correction [1, 3]. Effect sizes are reported using Cohen’s d [2].

Ablation setting	AUPR			EPR		
	P	Adjusted P	Effect size	P	Adjusted P	Effect size
w/o LEG	< 0.001	< 0.001	23.828	0.002	0.005	4.446
Random LEG	< 0.001	< 0.001	9.887	0.011	0.011	2.806
w/o GO	< 0.001	< 0.001	34.179	0.004	0.005	3.867
Random GO	< 0.001	< 0.001	12.337	0.002	0.004	4.346